# LLMs as Noisy Channels:
# A Shannon Perspective on Model Capacity and Scaling Laws

**Xu Ouyang**[* 1 2] **Deyi Liu**[2] **Yuhang Cai**[2 3] **Jing Liu**[2] **Yuan Yang**[2] **Chen Zheng**[2] **Thomas Hartvigsen**[1]
**Yiyuan Ma**[† 2]

## Abstract

Existing scaling laws for Large Language Models (LLMs), predominantly monotonic power laws, fail to explain emerging non-monotonic phenomena such as *catastrophic overtraining* and *quantization-induced degradation*, where performance deteriorates despite increased compute.

We propose the Shannon Scaling Law, a unified theoretical framework that models LLM training as information transmission over a noisy channel, grounded in the Shannon–Hartley theorem. By mapping model parameters to channel bandwidth and training tokens to signal power, our formulation explicitly captures the interaction between learning signal and intrinsic noise. This perspective reveals a fundamental Shannon capacity for LLMs: scaling model size or data without preserving a sufficient signal-to-noise ratio (SNR) inevitably amplifies noise, inducing a transition from monotonic improvement to U-shaped performance degradation.

We validate our theory through experiments on Pythia and OLMo2 under perturbations, including Gaussian noise, quantization and supervised fine-tuning on math, QA and code tasks. The Shannon Scaling Law consistently outperforms classical scaling laws and recent perturbation-aware laws, achieving strong $R^2$ scores and accurately capturing loss basins missed by prior approaches. It also extrapolates: fitted on $\leq$6.9B Pythia models with $\leq$180B tokens, it predicts the unseen 12B model up to 307B tokens at pooled $R^2$=0.847, while monotonic baselines collapse.

*Work done during internship at ByteDance Seed. †Project Manager. [1]Department of Computer Science, University of Virginia, USA [2]ByteDance Seed [3]University of California, Berkeley, USA. Correspondence to: Xu Ouyang <ftp8nr@virginia.edu>, Deyi Liu <deyiliu@bytedance.com>.

*Proceedings of the 43rd International Conference on Machine Learning*, Seoul, South Korea. PMLR 306, 2026. Copyright 2026 by the author(s).

## 1. Introduction

The prevailing paradigm in LLM development rests on the scaling hypothesis (Kaplan et al., 2020; Hoffmann et al., 2022): the empirical observation that model performance

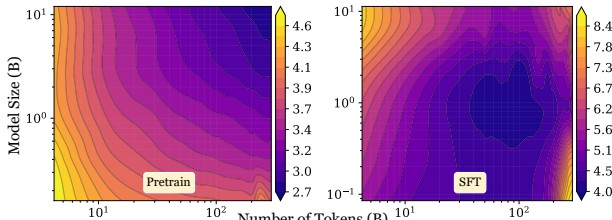

*Figure 1.* Loss landscapes between Pretraining and downstream SFT. While pretraining exhibits monotonic improvement, SFT reveals a loss basin, indicating that scaling either model size or token count beyond a critical threshold leads to performance degradation.

improves monotonically with increased compute, parameter count, and dataset size. This trajectory has driven the emergence of trillion-parameter Mixture-of-Experts models such as DeepSeek-V4 (1.6T) (DeepSeek-AI, 2026) and Kimi K2.6 (1T) (Kimi Team, 2026), along with massive pretraining corpora.

However, the assumption that "scaling is all you need" is facing practical challenges. Recent findings suggest that the scaling curve is not strictly monotonic and that naive scaling does not guarantee performance gains. Scaling laws have boundary conditions that we are beginning to encounter.

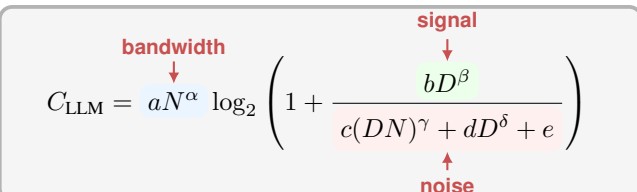

*Figure 2.* Our Shannon Scaling Law. $C_{\text{LLM}}$ denotes the capacity of LLMs. $N, D$ are model size and number of tokens. $\alpha, \beta, \gamma, \delta, a, b, c, d, e$ are fitted positive constants.

Specifically, Springer et al. (2025) challenge the monotonic belief by identifying *catastrophic overtraining*, where excessive pretraining degrades downstream fine-tuning performance. Similarly, Ouyang et al. (2024); Kumar et al. (2024)

observe that larger or more extensively trained models are paradoxically more susceptible to Quantization-induced Degradation (QiD). These phenomena produce U-shaped loss curves, where performance initially improves but eventually deteriorates. Traditional power-law formulations fail to model this trend effectively.

To address these anomalies, we propose a paradigm shift by viewing LLMs through the lens of communication systems (Shannon, 1948). We view a LLM as a noisy channel. In this analogy, the pretraining can be viewed as channel modulation (modulating information into model weights) and the inference is the transmission of information from input context $\mathcal{X}$ to output $\mathcal{Y}$. Just as physical channels are bounded, LLMs are bounded by noise from data and model architectures. Therefore, the Shannon-Hartley Theorem (Shannon, 1948), which defines the capacity of a noisy channel, offers a theoretical framework for LLM capacity.

Based on this perspective, we derive a new scaling law that defines the LLM's capacity $C_{\text{LLM}}$ analogous to the Shannon capacity in Figure 2. We map the components of the theorem to training dynamics as follows: (1) bandwidth corresponds to model size; (2) signal is derived from tokens; and (3) noise arises from three sources: data, model, and inevitable interference. During training, perturbations such as quantization introduce dynamic fluctuations captured by the noise term, reducing effective capacity and leading to the emergence of U-shaped scaling behaviors.

In this paper, we integrate power-law formulations of bandwidth, signal, and noise into the Shannon framework to propose a unified scaling law. Crucially, our framework reconciles the seemingly contradictory behaviors of monotonic scaling and U-shaped degradation. We posit that the strictly monotonic loss curves observed in standard pretraining represent a special case of the U-shaped phenomenon—specifically, a high-SNR regime where the perturbation factor is negligible. Our Shannon Scaling Law serves as a generalized formulation on both cases. In section 4, we demonstrate that this unified law outperforms existing baselines in various perturbation scenarios, including quantization (Frantar et al., 2023; Lin et al., 2024), SFT (Springer et al., 2025; Ouyang et al., 2022) and added Gaussian noise (Springer et al., 2025). More importantly, the law extrapolates: fitted on ≤6.9B Pythia models with ≤180B tokens, it predicts the unseen 12B model up to 307B tokens at pooled $R^2$=0.847, while OpenAI and Chinchilla collapse to negative scores (subsection 5.2).

## 2. Preliminary and Related Works

### 2.1. Scaling Laws for Large Language Models

Scaling laws quantify the relationship between model performance (loss or perplexity) and key factors such as model

parameters ($N$), training tokens ($D$).

**Monotonic Scaling Laws** Traditional works assume a strict power-law relationship where loss decreases monotonically as resources increase. OpenAI's scaling law (Kaplan et al., 2020) formulates the loss as:

$$L_{\text{OAI}}(N, D) = \left[ \left( \frac{N_c}{N} \right)^{\frac{\alpha_N}{\alpha_D}} + \frac{D_c}{D} \right]^{\alpha_D} \quad (1)$$

where $N_c, D_c$ are constant coefficients and $\alpha_N, \alpha_D$ are power-law exponents. Taking computational budget into account, Chinchilla law (Hoffmann et al., 2022) proposes an additive form fitted from optimal losses:

$$L_{\text{Chin}}(N, D) = \frac{A}{N^\alpha} + \frac{B}{D^\beta} + E \quad (2)$$

where $E$ represents the fitted irreducible loss, and $A, B, \alpha, \beta$ are fitted parameters.

**Perturbation-aware Scaling Laws** Recent studies challenge the monotonic assumption, identifying U-shaped loss curves caused by factors like quantization or overtraining (Springer et al., 2025; Ouyang et al., 2024). These laws typically introduce a degradation term to the base scaling law. Ouyang et al. (2024) model QiD by adding a penalty term $\Delta_q L$ to the OpenAI law:

$$L(N, D, P) = L_{\text{OAI}}(N, D) + \underbrace{k \cdot \frac{D^\beta}{N^\alpha P^\gamma}}_{\Delta_q L(N, D, P)} \quad (3)$$

where $P$ denotes quantization bit-width, $k$ is a fitted constant. Similarly, Kumar et al. (2024) propose an exponential degradation term $\delta_{\text{PTQ}}$ added to the Chinchilla law:

$$L(N, D, P) = L_{\text{Chin}}(N, D) + \underbrace{C_T \left( \frac{D^{\gamma_D}}{N^{\gamma_N}} \right) e^{-\frac{P}{\gamma}}}_{\delta_{\text{PTQ}}(N, D, P)} \quad (4)$$

where $C_T$ is a positive fitted constant. These formulations capture the trade-off where excessively large models or data sizes under low-bit precision lead to U-shaped loss curves.

### 2.2. Noisy Channel Modeling in Signal Processing

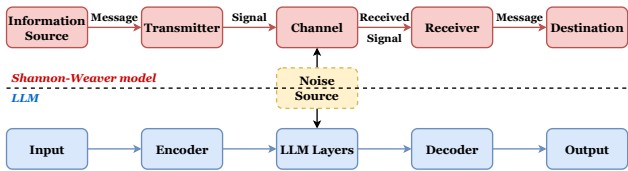

*Figure 3.* Structural correspondence between Shannon communication model (Shannon, 1948) and LLMs.

**The Shannon-Weaver Model and LLMs** The Shannon-Weaver model (Shannon, 1948; Haykin, 2001) describes communication as a linear process: *Source → Transmitter → Channel* (with Noise) *→ Receiver → Destination* (Figure 3). Besides, prior deep learning works (Shwartz-Ziv & Tishby, 2017; Tishby & Zaslavsky, 2015) have characterized DNNs through the mutual information $I(\mathcal{X}; \mathcal{Y})$ between the input $\mathcal{X}$ and the output $\mathcal{Y}$. $I(\mathcal{X}; \mathcal{Y})$ is also one of the theoretical foundations of the Shannon-Weaver model. Hence, we propose our law based on the similarities between this model and LLMs.

**Shannon-Hartley Theorem (Shannon, 1948)** This theorem defines the channel capacity $C$. This is the theoretical upper bound on the information rate for error-free transmission over a channel with bandwidth $B$ and additive white Gaussian noise (AWGN):

$$C = B \log_2 \left( 1 + \frac{S}{\mathcal{N}} \right) \quad (5)$$

Here, $S$ represents the average signal power, $\mathcal{N}$ is the noise power, and $S/\mathcal{N}$ is the Signal-to-Noise Ratio (SNR). In our work, we reinterpret these physical quantities to model the representational capacity of LLMs.

**Noisy Channel Model in NLP** The adoption of the Noisy Channel Model is well-established in NLP (Jurafsky & Martin, 2025), particularly in tasks such as spelling correction (Brill & Moore, 2000) and machine translation (Brown et al., 1993). These traditional approaches typically rely on Bayes' theorem to maximize the posterior probability[1]. However, our work fundamentally differs from this paradigm. Instead of using the channel model for the probabilistic inference of text sequences, we leverage this model to quantify the capacity of LLMs.

## 3. The Shannon Scaling Law

Inspired by Shannon capacity (Shannon, 1948; Haykin, 2001), we propose a novel scaling law that conceptualizes LLMs as a noisy channel. We define the model's capability $C_{\text{LLM}}$, which is analogous to channel capacity, as the upper bound on the rate at which knowledge can be learned and represented given a specific compute and data budget.

### 3.1. The Formulation of Shannon Scaling Law

We extend the channel capacity (Equation 5) to LLMs by mapping the physical components $(B, S, \mathcal{N})$ to model sizes $(N)$ and training tokens $(D)$. The proposed **Shannon Scal-**

---

[1] https://web.stanford.edu/~jurafsky/slp3/slides/6_Spell.pdf

**ing Law** is formulated as:

$$C_{\text{LLM}} = aN^\alpha \log_2 \left( 1 + \frac{bD^\beta}{c(DN)^\gamma + dD^\delta + e} \right) \quad (6)$$

where $a, b, c, d, e, \alpha, \beta, \gamma, \delta$ are fitted positive constants.

### 3.2. Component Analysis

**Bandwidth** In communication systems, bandwidth $B$ defines the range of frequencies available for transmission (Haykin, 2001): a wider channel allows more information throughput. Analogously, we believe that the model size ($N$) acts as the bandwidth of an LLM. Larger models possess a larger space, allowing them to capture a broader spectrum of features and patterns. Following established scaling conventions, we model this relationship as a power law:

$$B_{\text{LLM}} \propto N^\alpha \quad (7)$$

**Signal** A signal is a function that conveys information about the behavior of a system or attributes of some phenomenon (Priemer, 1990). For LLMs, the "signal" is the knowledge embedded within the training corpus ($D$). Established scaling conventions model the information gain from larger $D$ as a power law. Assuming the training data is sampled from a vast, information-rich distribution, the average signal power is proportional to the number of training tokens. Thus, we define the signal power as:

$$S_{\text{LLM}} \propto D^\beta \quad (8)$$

**Noise** Noise represents unwanted perturbations that degrade the signal. In the context of LLM training, noise is inevitable and we believe they arise from two distinct sources, which our formulation explicitly captures:

- **Data-Induced Noise** ($dD^\delta$): Data inevitably contains noise (e.g., typos, ambiguities, and contradictions). As the token count $D$ increases, the model becomes increasingly sensitive to such noise (Ouyang et al., 2024; Springer et al., 2025). Given that $D = bs \times t$, where $bs$ denotes the batch size and $t$ the training steps. This term effectively captures the accumulation of data-induced noise throughout the training trajectory.
- **Model-Interaction Noise** ($c(DN)^\gamma$): The training process can be viewed as a denoising procedure (Vincent et al., 2008; Shwartz-Ziv & Tishby, 2017): randomly initialized models having substantial noise and very low capacity $C$, and are progressively denoised as the training step $t$ increases. This term models this dynamic duration. Since $D$ is proportional to $t$, this term tracks the intrinsic model noise changing over the training trajectory $t$.
- **Irreducible Noise** ($e$): A constant term representing irreducible system entropy, such as architectural limita-

tions. This is analogous to the fitted constant $E$ found in the Chinchilla scaling laws (Hoffmann et al., 2022).

## 3.3. Linking Capacity to Loss

For LLMs, loss or perplexity corresponds to the "error rate". We propose a reciprocal relationship between test loss $\mathcal{L}$ and model capacity $C_{\text{LLM}}$:

$$\mathcal{L}(N, D) = \frac{1}{C_{\text{LLM}}} \qquad (9)$$

This formulation satisfies our two principles:

1. As capacity approaches infinity ($C \to \infty$), loss approaches 0 ($\mathcal{L} \to 0$). Conversely, a zero-capacity channel results in very large loss ($\mathcal{L} \to \infty$).
2. Nonlinearity. At high loss values (early training), small capacity gains yield significant loss reductions. However, as the model converges, achieving marginal loss reduction requires much larger increases in capacity.

## 4. Experiments

In this section, we conduct extensive experiments to validate the effectiveness and universality of the proposed Shannon Scaling Law across various model architectures, datasets and perturbation sources.

**Scaling Law Baselines** As discussed in section 2, we benchmark our method against several scaling laws, including those designed to model the monotonic trend and overtraining phenomenon.

| Method | Law Formulation |
|---|---|
| *Power Laws* | |
| OpenAI Law (Kaplan et al., 2020) | $L(N, D) = \left[ \left( \frac{a}{N} \right)^{\frac{\alpha}{\beta}} + \frac{b}{D} \right]^{\beta}$ |
| Chinchilla Law (Hoffmann et al., 2022) | $L(N, D) = \frac{a}{N^\alpha} + \frac{b}{D^\beta} + c$ |
| *Perturbation-aware Laws* | |
| QiD Law (Ouyang et al., 2024) | $L(N, D, X) = \frac{a}{N^\alpha} + \frac{b}{D^\beta} + c + dN^{\alpha'} D^{\beta'} X^\gamma$ |
| Law of Precision (Kumar et al., 2024) | $L(N, D, X) = \frac{a}{N^\alpha} + \frac{b}{D^\beta} + c + dN^{\alpha'} D^{\beta'} e^{\gamma X}$ |
| Symmetric Law | $L(N, D) = a\frac{N^\alpha}{D^\beta} + b\frac{D^\beta}{N^\alpha} + c$ |
| Asymmetric Law | $L(N, D) = a\frac{N^\alpha}{D^\beta} + b\frac{D^{\beta'}}{N^{\alpha'}} + c$ |

*Table 1.* Overview of scaling laws. Beyond established Power Laws and Perturbation-aware methods, we introduce Symmetric and Asymmetric Laws, our extensions derived from the Chinchilla law on the non-monotonic phenomenon.

**Models, Datasets and Perturbation Sources** We primarily utilize two open-source model suites: Pythia (Biderman et al., 2023) and OLMo2 (OLMo et al., 2025), both of which provide intermediate checkpoints across various scales. For Pythia, we use the Pythia-dedup suite trained on the deduplicated Pile (Gao et al., 2020) dataset for 1.5 epochs, covering six sizes: 160M, 410M, 1B, 2.8B, 6.9B, and 12B. For OLMo2 series. We use the 1B, 7B, 13B and 32B models. To ensure consistency with the training stage of Pythia suite, we only use stage-1 checkpoints.

We use the test loss of above models on the wikitext2 (Merity et al., 2016) dataset as the law fitting target values. There are three perturbation sources that we investigate: Gaussian noise, SFT and quantization. Following the SFT protocol in Springer et al. (2025), we perform full fine-tuning on three tasks: GSM8K (Cobbe et al., 2021) (Math), SiQA (Sap et al., 2019) (QA), and StarCoder-Python (Li et al., 2023) (Coding), with identical hyperparameters. Finally, we quantize the model checkpoints from 16 bit to 2 bit, 3 bit and 4 bit using GPTQ (Frantar et al., 2023). The perturbed checkpoints are subsequently evaluated on wikitext2. Please refer to Appendix A for implementation details.

### 4.1. Gaussian Noise as Perturbation

Springer et al. (2025) observed a phenomenon termed *progressive sensitivity to noise*: for a fixed perturbation magnitude, the degradation in perplexity increases monotonically with the number of training tokens. Larger perturbations lead to sharper degradation, causing the inflection point to occur at lower token budgets.

We followed their approach and modify the noise injection strategy slightly. Instead of scaling by the initialization covariance matrix $\Sigma$, we inject additive Gaussian noise $\epsilon \sim \mathcal{N}(0, \sigma_n^2)$ based on the Signal-to-Noise Ratio (SNR). This decision was made for two reasons: 1. we found this phenomenon was difficult to reproduce consistently using their covariance method; 2. the initialization checkpoints are not available for all open-source models. It is feasible for them as they train their own closed-source models.

To get each perturbed weight $\tilde{w}$, we add noise $n$ to weight $w$ by generating noise with variance $\sigma_n^2$ based on the power of weight $P_w$ and the target $SNR_{dB}$ in decibels (dB) (Gonzalez & Woods, 2008; Oppenheim & Schafer, 2009):

$$\tilde{w} = w + n, \quad \text{where } n \sim \mathcal{N}(0, \sigma_n^2) \qquad (10)$$

$$\sigma_n^2 = \frac{P_w}{SNR} = \frac{\mathbb{E}[|w|^2]}{10^{SNR_{dB}/10}} \qquad (11)$$

This approach allows us to strictly control the perturbation power relative to the weights power across all weights.

**Emergence of U-shaped Curves in Loss Landscapes under Gaussian Noise** The evolution of the loss landscape in Figure 4, from left (low noise) to right (high noise), reveals a fundamental shift in scaling dynamics.

In the high SNR regime, the loss landscape follows traditional scaling laws. The loss contours are open, indicating that increasing either model size or training tokens monotonically reduces loss. However, as the noise level increases (with SNR decreasing to 30 dB and 20 dB), this monotonicity breaks. In the bottom-right corner, a high-loss region becomes increasingly prominent. U-shaped curves emerge along both axes: (1) for a fixed model size, increasing tokens

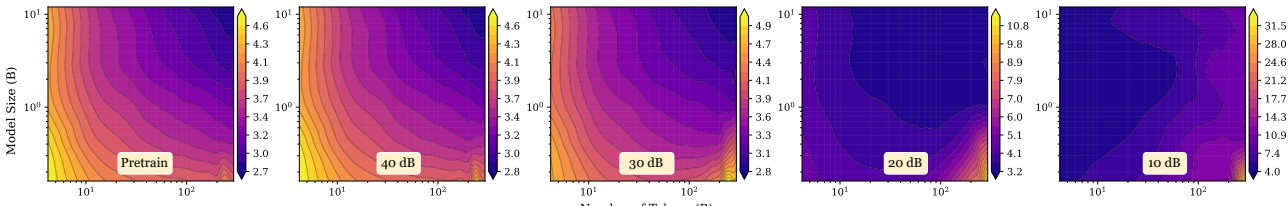

*Figure 4.* Evolution of Pythia loss contours under varying Gaussian noise levels. The emergence of a loss basin at higher noise levels.

| Gaussian Noise | Pythia | | | | | | | OLMo2 | | | | |
|---|---|---|---|---|---|---|---|---|---|---|---|---|
| $\mathcal{L}(N,D)/\mathcal{L}(N,D,X)$ | 40 dB | 30 dB | 20 dB | 15 dB | 12 dB | 10 dB | Avg ± Std | 40 dB | 30 dB | 20 dB | 10 dB | Avg ± Std |
| $\frac{1}{aN^\alpha \log_2\left(1+\frac{bD^\beta}{c(DN)^\gamma+dD^\delta+e}\right)}$ | **0.9895** | **0.9882** | **0.9672** | **0.9442** | **0.9234** | **0.9555** | **0.9613 ± 0.03** | **0.9830** | **0.9907** | **0.9908** | 0.8695 | **0.9585 ± 0.06** |
| $\left[\left(\frac{a}{N}\right)^{\frac{\alpha}{\beta}}+\frac{b}{D}\right]^\beta$ | 0.9786 | 0.8951 | 0.4865 | 0.3265 | 0.1750 | 0.0707 | 0.4887 ± 0.38 | 0.9293 | 0.9744 | 0.9884 | 0.5201 | 0.8531 ± 0.22 |
| $\frac{a}{N^\alpha}+\frac{b}{D^\beta}+c$ | 0.9523 | 0.8458 | 0.4749 | 0.3762 | 0.2844 | 0.2395 | 0.5289 ± 0.30 | 0.9517 | 0.9756 | 0.9801 | 0.5201 | 0.8569 ± 0.22 |
| $\frac{a}{N^\alpha}+\frac{b}{D^\beta}+c+d\frac{D^{\beta'}}{N^{\alpha'}X^\gamma}$ | 0.9891 | 0.9820 | 0.9671 | 0.9417 | 0.8759 | 0.8251 | 0.9302 ± 0.07 | 0.9535 | 0.9805 | 0.9850 | **0.8708** | 0.9475 ± 0.05 |
| $\frac{a}{N^\alpha}+\frac{b}{D^\beta}+c+d\frac{D^{\beta'}}{N^{\alpha'}e^{\gamma X}}$ | 0.9523 | 0.8458 | 0.9671 | 0.9417 | 0.8759 | 0.8251 | 0.9013 ± 0.06 | 0.9517 | 0.9756 | 0.9801 | 0.8705 | 0.9445 ± 0.05 |
| $a\frac{N^\alpha}{D^\beta}+b\frac{D^\beta}{N^\alpha}+c$ | 0.9837 | 0.9360 | 0.9151 | 0.9342 | 0.8681 | 0.8251 | 0.9103 ± 0.06 | 0.9502 | 0.9786 | 0.9768 | 0.8419 | 0.9369 ± 0.06 |
| $a\frac{N^\alpha}{D^\beta}+b\frac{D^{\beta'}}{N^{\alpha'}}+c$ | 0.9879 | 0.9729 | 0.9643 | 0.9380 | 0.8681 | 0.8322 | 0.9272 ± 0.06 | 0.9533 | 0.9851 | 0.9904 | 0.8568 | 0.9464 ± 0.06 |

*Table 2.* Comparison of $R^2$ scores under varying Gaussian noise levels. Our Shannon Scaling Law (Row 1) demonstrates superior robustness across the full spectrum (40 dB–10 dB).

initially reduces loss but eventually leads to degradation, visible as the color shifts from blue (low loss) back to yellow (high loss) on the far right; (2) similarly, for a fixed token budget, increasing model size beyond a certain threshold causes the loss to rise. This validates our observation that excessively large models amplify their model noise when the signal is insufficient. Under the extreme 10 dB condition, the region of low loss shrinks significantly and the overall loss values increase drastically. This shows that when noise dominates the channel, simply scaling up $D$ is detrimental.

**Fitting $\mathcal{L}(N,D)$ under Varying Noise Levels**   Table 2 presents a comparative analysis of goodness-of-fit, quantified by the $R^2$ score, across varying levels of Gaussian noise (10 dB – 40 dB) for both Pythia (Biderman et al., 2023) and OLMo2 (OLMo et al., 2025) model series. The results demonstrate that our proposed law (Row 1) consistently outperforms baseline laws across diverse perturbation levels. Our law achieves the highest alignment with experimental results across all tested noise levels. In low-noise results (40 dB), our model achieves the highest scores of 0.9895 for Pythia and 0.9830 for OLMo2. Notably, as the noise increases, the performance gap between our method and the baselines widens. At the highest noise level (10 dB), our model maintains a robust $R^2$ of 0.9555 on Pythia, significantly surpassing the next best-performing baseline (Asymmetric law, Row 7) drops to 0.8322. Similarly, on OLMo2, our method retains a competitive score of 0.8695.

The stability of our approach is evidenced by the "Average ± Standard Deviation" column. Our method yields robust average $R^2$ of 0.9613 ± 0.03 (Pythia) and 0.9585 ± 0.06 (OLMo2), indicating not only high scores but also exceptional consistency. In contrast, competing laws exhibit significant volatility. For instance, the OpenAI law shows high standard deviations of ±0.38 and ±0.22, reflecting its inability to model the data effectively as the SNR decreases. While perturbation-aware baselines (Rows 4–7) struggle to maintain both consistency and accuracy across the full spectrum of noise levels. The Shannon capacity structure of our proposed formulation effectively models the loss landscapes. This makes ours the only law to consistently achieve an average $R^2 > 0.95$ across both model families.

### 4.2. Supervised Fine-Tuning as Perturbation

**Emergence of U-shaped Curves in Loss Landscapes under SFT**   We perform full fine-tuning on all the datasets and pretraining checkpoints with the same hyperparameters, except learning rate (LR). In the low-LR regime (left) of Figure 5, the landscape exhibits classic monotonic scaling, where increasing model size ($N$) or tokens ($D$) consistently reduces loss. However, as LR rises, the landscape distorts. Similar to the Gaussian perturbation results, we observe the emergence of U-shaped curves. Crucially, a "basin" of loss emerges at the center (LR=2e-4). This shows U-shaped trends along both the $N$ and $D$ axes: for a fixed token or size

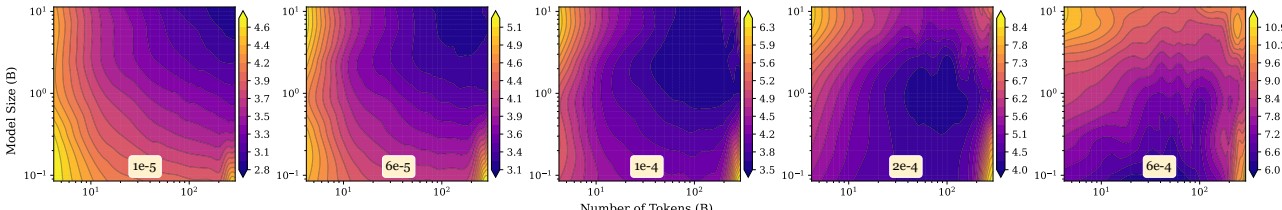

*Figure 5.* Pythia loss evolution under varying SFT learning rates on GSM8K. The emergence of a loss basin at higher noise levels.

| SFT | GSM8K | | | | | | SiQA | | | | | | StarCoder | | | | | |
|---|---|---|---|---|---|---|---|---|---|---|---|---|---|---|---|---|---|---|
| $\mathcal{L}(N,D)/\mathcal{L}(N,D,X)$ | 1e-5 | 6e-5 | 1e-4 | 2e-4 | 6e-4 | Avg ± Std | 1e-5 | 6e-5 | 1e-4 | 2e-4 | 6e-4 | Avg ± Std | 1e-5 | 6e-5 | 1e-4 | 2e-4 | 6e-4 | Avg ± Std |
| $\frac{1}{aN^\alpha \log_2\left(1+\frac{bD^\beta}{c(DN)^\gamma+dD^\delta+e}\right)}$ | **0.986** | **0.970** | **0.901** | **0.874** | **0.950** | **0.936 ± 0.05** | **0.987** | **0.924** | **0.831** | **0.920** | **0.921** | **0.916 ± 0.06** | 0.977 | **0.976** | **0.973** | 0.901 | **0.858** | **0.937 ± 0.05** |
| $\left[\left(\frac{a}{N}\right)^{\frac{\alpha}{\beta}}+\frac{b}{D}\right]^\beta$ | 0.963 | 0.847 | 0.466 | 0.000 | -0.579 | 0.339 ± 0.64 | 0.956 | 0.580 | 0.000 | 0.000 | -0.879 | 0.131 ± 0.70 | 0.956 | 0.942 | 0.881 | 0.507 | -8.336 | -1.010 ± 4.10 |
| $\frac{a}{N^\alpha}+\frac{b}{D^\beta}+c$ | 0.936 | 0.813 | 0.477 | 0.020 | 0.000 | 0.449 ± 0.44 | 0.928 | 0.580 | 0.167 | 0.014 | 0.007 | 0.339 ± 0.40 | 0.928 | 0.903 | 0.832 | 0.507 | 0.002 | 0.634 ± 0.39 |
| $\frac{a}{N^\alpha}+\frac{b}{D^\beta}+c+d\frac{D^{\beta'}X^\gamma}{N^{\alpha'}}$ | 0.934 | 0.814 | 0.552 | 0.576 | 0.308 | 0.637 ± 0.24 | 0.929 | 0.648 | 0.379 | 0.488 | 0.427 | 0.574 ± 0.22 | 0.929 | 0.899 | 0.844 | 0.605 | 0.384 | 0.732 ± 0.23 |
| $\frac{a}{N^\alpha}+\frac{b}{D^\beta}+c+d\frac{D^{\beta'}e^{\gamma X}}{N^{\alpha'}}$ | 0.984 | 0.949 | 0.851 | 0.572 | 0.308 | 0.733 ± 0.29 | 0.985 | 0.868 | 0.748 | 0.488 | 0.223 | 0.662 ± 0.31 | **0.980** | 0.974 | 0.961 | **0.913** | 0.384 | 0.842 ± 0.26 |
| $a\frac{N^\alpha}{D^\beta}+b\frac{D^\beta}{N^\alpha}+c$ | 0.970 | 0.892 | 0.684 | 0.659 | 0.802 | 0.802 ± 0.13 | 0.965 | 0.735 | 0.663 | 0.683 | 0.757 | 0.760 ± 0.12 | 0.963 | 0.950 | 0.904 | 0.677 | 0.641 | 0.827 ± 0.16 |
| $a\frac{N^\alpha}{D^\beta}+b\frac{D^{\beta'}}{N^{\alpha'}}+c$ | 0.978 | 0.933 | 0.819 | 0.804 | 0.944 | 0.896 ± 0.08 | 0.974 | 0.838 | 0.758 | 0.886 | 0.920 | 0.875 ± 0.08 | 0.970 | 0.963 | 0.941 | 0.872 | 0.832 | 0.916 ± 0.06 |

*Table 3.* Comparison of $R^2$ scores across diverse SFT domains: GSM8K, SiQA and StarCoder. Our Shannon Scaling Law demonstrates universal applicability, consistently achieving the highest average scores: 0.936, 0.916 and 0.937.

budget, excessively large models or overtraining begin to exhibit performance degradation. At the highest LR (right), the system undergoes *catastrophic overtraining* (Springer et al., 2025): the low-loss region virtually disappears, replaced by high loss values across the board. Neither scaling up the model nor adding more tokens can compensate for the destructive interference. This mirrors the "capacity collapse" predicted by our Shannon Law when the noise term dominates the denominator. Please refer to Appendix A for more loss contour plots on SiQA and StarCoder. Such "loss basins" also clearly exist on these two datasets.

**Fitting $\mathcal{L}(N,D)$ across Varying Learning Rates on Diverse SFT Datasets**  Table 3 extends our evaluation to three distinct SFT tasks: GSM8K (Merity et al., 2016), SiQA (Sap et al., 2019), and StarCoder (Li et al., 2023), under varying learning rates. Different from precision-based scaling (Ouyang et al., 2024; Kumar et al., 2024), the learning rate $X$ acts inversely: a larger $X$ induces greater perturbation. We places the $X^\gamma$ term in the numerator to correctly model this noise amplification.

Due to the lack of perturbation term, OpenAI and Chinchilla laws (Rows 2–3) exhibit catastrophic failure, even yielding negative $R^2$ values (e.g., $-1.010$ avg for OpenAI on StarCoder). Comparing against perturbation-aware baselines (Rows 4–7), our method demonstrates ubiquitous superiority. Even the strongest competitor (Row 7) consistently underperforms our Shannon Scaling Law across all datasets:

0.936 vs. 0.896 on GSM8K, 0.916 vs. 0.875 on SiQA, and 0.937 vs. 0.916 on StarCoder. Crucially, our advantage is most pronounced within the "loss basins." For instance, at LR = 1e−4 and 2e−4, our law maintains robust fits of 0.901 and 0.874 on GSM8K, significantly outperforming the best baseline (0.819 and 0.804). This trend extends to SiQA and StarCoder. Such consistent superiority across diverse SFT datasets and perturbation levels confirms our formulation as the universally superior predictor for SFT perturbation dynamics.

### 4.3. Quantization as Perturbation

**Emergence of U-shaped Curves in Loss Landscapes under Quantization**  Figure 6 illustrates the evolution of loss contours under post-training quantization GPTQ (Frantar et al., 2023). At 4 bit precision, the landscape retains standard monotonic scaling, indicating sufficient fidelity where increasing $N$ and $D$ yields consistent gains. However, as precision drops, the landscape distorts, quantization noise dominates, causing the region of optimal loss to collapse into a confined "basin" rather than an open expanse.

**Fitting $\mathcal{L}(N,D)$ across Different Precisions**  To validate architectural universality, we evaluate the scaling laws on both Pythia and OLMo2 suites under varying quantization levels. Table 4 confirms that our proposed law consistently yields the highest fitting scores, achieving averages of 0.9824 (Pythia) and 0.9548 (OLMo2). The superiority is most pronounced in the extreme 2-bit regime, where stan-

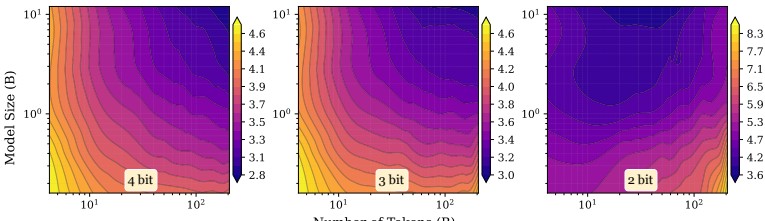

*Figure 6.* Evolution of Pythia loss contours under varying quantization bit-widths. The emergence of a loss basin at higher noise levels.

| Quantization | Pythia | | | | OLMo2 | | | |
|---|---|---|---|---|---|---|---|---|
| $\mathcal{L}(N,D)/\mathcal{L}(N,D,X)$ | 4 bit | 3 bit | 2 bit | Avg ± Std | 4 bit | 3 bit | 2 bit | Avg ± Std |
| $\frac{1}{aN^\alpha \log_2\left(1+\frac{bD^\beta}{c(DN)^\gamma+dD^\delta+e}\right)}$ | **0.9953** | **0.9917** | **0.9602** | **0.9824 ± 0.02** | **0.9886** | **0.9888** | **0.8869** | **0.9548 ± 0.06** |
| $\left[\left(\frac{a}{N}\right)^{\frac{\alpha}{\beta}}+\frac{b}{D}\right]^\beta$ | 0.9893 | 0.9817 | 0.7201 | 0.8970 ± 0.15 | 0.9766 | 0.9871 | 0.6155 | 0.8597 ± 0.21 |
| $\frac{a}{N^\alpha}+\frac{b}{D^\beta}+c$ | 0.9770 | 0.9688 | 0.7312 | 0.8923 ± 0.14 | 0.9732 | 0.9700 | 0.6155 | 0.8529 ± 0.21 |
| $\frac{a}{N^\alpha}+\frac{b}{D^\beta}+c+d\frac{D^{\beta'}}{N^{\alpha'}X^\gamma}$ | 0.9932 | 0.9883 | 0.9549 | 0.9788 ± 0.02 | 0.9790 | 0.9791 | 0.8353 | 0.9311 ± 0.08 |
| $\frac{a}{N^\alpha}+\frac{b}{D^\beta}+c+d\frac{D^{\beta'}}{N^{\alpha'}e^{\gamma X}}$ | 0.9932 | 0.9883 | 0.9549 | 0.9788 ± 0.02 | 0.9790 | 0.9791 | 0.8353 | 0.9311 ± 0.08 |
| $a\frac{N^\alpha}{D^\beta}+b\frac{D^\beta}{N^\alpha}+c$ | 0.9833 | 0.9735 | 0.9163 | 0.9577 ± 0.04 | 0.9789 | 0.9745 | 0.8713 | 0.9416 ± 0.06 |
| $a\frac{N^\alpha}{D^\beta}+b\frac{D^{\beta'}}{N^{\alpha'}}+c$ | **0.9953** | 0.9895 | 0.9499 | 0.9782 ± 0.02 | 0.9850 | 0.9882 | 0.8868 | 0.9533 ± 0.06 |

*Table 4.* Comparison of $R^2$ scores across different quantization bit-widths: 4, 3 and 2 bit. Our Shannon Scaling Law consistently outperforms other laws.

dard power laws suffer catastrophic collapse: dropping to $R^2 \approx 0.72$ for Pythia and plummeting further to $\approx 0.61$ for OLMo2. In contrast, our method maintains robust fidelity (0.9602 and 0.8869, respectively). This superiority validates that our formulation effectively parameterizes the non-monotonic saturation dynamics caused by lower precision, distinguishing it as the only law to consistently exceed $R^2 > 0.96$ with Pythia even under aggressive quantization.

### 4.4. Pretraining-only Trajectories as a High-SNR Special Case

Unperturbed trajectories can be viewed as a high-SNR special case. Our law achieves good precision on original pre-training loss, yielding an $R^2$ of 0.9889 on OLMo2, which outperforms OpenAI law's 0.9713, Chinchilla law's 0.9737, and Asymmetric law's 0.9833. On Pythia, our law reaches 0.9915, surpassing Chinchilla law's 0.9681 and OpenAI law's 0.9848 while matching Asymmetric law's 0.9921. Beyond the performance, the decisive advantage of our law is its universality: while standard baselines collapse under external noise, the Shannon Scaling Law uniquely unifies high-SNR pretraining and low-SNR perturbed regimes within a single, robust theoretical formulation.

## 5. Beyond Fitting: Extrapolation and Analysis

### 5.1. Towards Parameter-Efficient Scaling Laws

The full Shannon Scaling Law needs 9 fitted constants, which is the same as QiD law (Ouyang et al., 2024) and Law

of precision (Kumar et al., 2024). However, other baseline laws need fewer fitted constant.

To enhance parameter efficiency, we derive a Simplified Shannon Law with only 6 parameters $(a, c, \alpha, \beta, \gamma, \delta)$ by pruning those consistently fitted to negligible values:

$$C_{\text{Simpl}} = aN^\alpha \log_2\left(1 + \frac{D^\beta}{c(DN)^\gamma + D^\delta}\right) \quad (12)$$

The left panel of Table 6 confirms that the simplified law retains near-full predictive power (Average $R^2$: 0.9541 vs. 0.9656). Crucially, it also outperforms the best baselines on different Gaussian noise levels, particularly in high-noise regimes (e.g., 0.9092 vs. 0.8322 at 10 dB). This establishes that even in its reduced form, our law offers a superior trade-off between complexity and fitting accuracy.

### 5.2. Predictive Power via Extrapolation to Unseen Models and Tokens

The fitting results above quantify how well each law explains observed data, but the practical value of a scaling law lies in its ability to *predict* the behaviour of *larger, unseen* models trained on *more, unseen* tokens. To test this, we run three extrapolation experiments on the Pythia suite under Gaussian noise. All reported scores are **pooled** $R^2$ computed by concatenating held-out predictions across all six SNR levels (10–40 dB) into a single flat array. This is strictly stricter than averaging per-SNR $R^2$ and rules out per-level cherry-picking.

| Exponents | | Pretrain | GSM8K | | | | | | SiQA | | | | | | StarCoder | | | | | |
|---|---|---|---|---|---|---|---|---|---|---|---|---|---|---|---|---|---|---|---|---|
| | | | 1e-5 | 3e-5 | 6e-5 | 1e-4 | 2e-4 | 1e-3 | 1e-5 | 3e-5 | 6e-5 | 1e-4 | 2e-4 | 1e-3 | 1e-5 | 3e-5 | 6e-5 | 1e-4 | 2e-4 | 1e-3 |
| $N$ | Bandwidth $\alpha$ | **0.302** | **0.706** | **0.640** | 0.496 | 0.321 | 0.280 | 0.286 | **0.780** | **0.567** | 0.338 | 0.244 | 0.335 | 0.642 | **0.659** | **0.661** | **0.661** | **0.706** | 0.471 | 0.318 |
| | Noise $\gamma$ | 0.299 | 0.676 | 0.620 | **0.505** | **0.475** | **0.554** | **0.357** | 0.748 | 0.558 | **0.436** | **0.483** | **0.548** | **0.684** | 0.632 | 0.635 | 0.637 | 0.688 | **0.515** | **0.439** |
| $D$ | Signal $\beta$ | 0.402 | 0.742 | 0.683 | 0.577 | 0.579 | 0.602 | 0.409 | 0.814 | 0.618 | 0.521 | 0.605 | 0.617 | 0.696 | 0.707 | 0.710 | 0.712 | 0.762 | 0.609 | 0.431 |
| | Noise $\delta$ | **0.745** | 2.600 | 3.363 | 3.557 | 4.297 | 3.296 | 1.676 | 3.078 | 3.955 | 4.073 | 3.367 | 2.724 | 3.082 | 2.312 | 2.498 | 2.898 | 3.836 | 3.685 | 2.913 |

*Table 5.* Comparison of exponents for model size ($N$) and token count ($D$). The relationship between bandwidth exponent ($\alpha$) and model noise exponent ($\gamma$) inverts as perturbation intensity increases. In contrast, the data noise exponent ($\delta$) persistently overshadows the signal exponent ($\beta$) across all levels of perturbations.

| Gaussian Noise | Method | Pretrain | 40 dB | 30 dB | 20 dB | 15 dB | 12 dB | 10 dB | Avg ± Std | Method | Pretrain | 40 dB | 30 dB | 20 dB | 15 dB | 12 dB | 10 dB | Avg ± Std |
|---|---|---|---|---|---|---|---|---|---|---|---|---|---|---|---|---|---|---|
| **Pythia** | Full | 0.9915 | **0.9895** | **0.9882** | 0.9672 | 0.9442 | 0.9234 | 0.9555 | **0.9656 ± 0.03** | $c(DN)^\gamma$ | 0.9915 | **0.9895** | 0.9882 | 0.9672 | 0.9442 | **0.9234** | 0.9555 | **0.9656 ± 0.03** |
| | Simplified | 0.9880 | 0.9889 | 0.9857 | 0.9653 | 0.9321 | 0.9095 | 0.9092 | 0.9541 ± 0.04 | $cN^\gamma$ | | **0.9926** | **0.9895** | **0.9896** | **0.9726** | **0.9472** | 0.8908 | 0.8035 | 0.9408 ± 0.07 |
| | Pareto Front | **0.9921** | 0.9891 | 0.9820 | 0.9671 | 0.9417 | 0.8759 | 0.8322 | 0.9400 ± 0.06 | Pareto Front | 0.9921 | 0.9891 | 0.9820 | 0.9671 | 0.9417 | 0.8759 | 0.8322 | 0.9400 ± 0.06 |
| **OLMo2** | Full | **0.9889** | **0.9830** | **0.9907** | **0.9908** | - | - | 0.8695 | **0.9646 ± 0.05** | $c(DN)^\gamma$ | **0.9889** | **0.9830** | **0.9907** | **0.9908** | - | - | 0.8695 | **0.9646 ± 0.05** |
| | Simplified | 0.9877 | 0.9779 | 0.9895 | 0.9903 | - | - | 0.8695 | 0.9630 ± 0.05 | $cN^\gamma$ | 0.9879 | 0.9776 | 0.9896 | 0.9903 | - | - | 0.8695 | 0.9630 ± 0.05 |
| | Pareto Front | 0.9833 | 0.9535 | 0.9851 | 0.9904 | - | - | **0.8708** | 0.9566 ± 0.05 | Pareto Front | 0.9833 | 0.9535 | 0.9851 | 0.9904 | - | - | **0.8708** | 0.9566 ± 0.05 |

*Table 6.* Evaluation of law variants. *Left*: Comparison of fitting accuracy between our full law, its simplified 6-parameter variant, and the best-performing baselines (Pareto Front). *Right*: Ablation analysis of the model-noise term. The significant gap at 10 dB validates the necessity of $c(DN)^\gamma$ over the size-only $cN^\gamma$ as the model noise term.

| Method | $j=8$ Train $\leq$75.5B Predict 75.5B–307B | $j=12$ Train $\leq$180.4B Predict 180.4B–307B | $j=15$ Train $\leq$272.6B Predict 272.6B–307B |
|---|---|---|---|
| Shannon (Ours, 9p) | 0.611 | **0.781** | 0.941 |
| Shannon-Simpl (Ours, 6p) | 0.768 | 0.753 | **0.945** |
| Chinchilla | 0.197 | 0.265 | 0.352 |
| OpenAI | 0.162 | 0.245 | 0.297 |
| QiD Law | 0.441 | 0.766 | 0.862 |
| Law of Precision | 0.486 | 0.632 | 0.743 |
| Symmetric Law | 0.726 | 0.760 | 0.860 |
| Asymmetric Law | **0.805** | 0.774 | 0.851 |

*Table 7.* Progressive Token Extrapolation on Pythia. We fit each law on the first $j$ token checkpoints (out of 16 spanning up to 307B tokens) and report pooled $R^2$ over all remaining points concatenated across 6 SNR levels.

**Token Extrapolation** We fit each law on the first $j$ token checkpoints of every model (out of 16 checkpoints spanning up to 307B tokens) and predict the remaining ones. As shown in Table 7, the Shannon Scaling Law variants consistently dominate the perturbation-aware baselines at every $j$. Notably, the 6-parameter Shannon-Simpl reaches $R^2 = 0.945$ at $j=15$, exceeding QiD Law's $0.862$ and Law of Precision's $0.743$ despite using fewer fitted constants.

| Method | $k=3$ Train [1B, 410M, 160M] Predict [12B, 6.9B, 2.8B] | $k=4$ Train $\leq$2.8B Predict [12B, 6.9B] | $k=5$ Train $\leq$6.9B Predict [12B] |
|---|---|---|---|
| Shannon (Ours, 9p) | 0.521 | 0.787 | 0.845 |
| Shannon-Simpl (Ours, 6p) | **0.605** | **0.837** | **0.847** |
| Chinchilla | 0.261 | 0.726 | 0.702 |
| OpenAI | −0.507 | −0.048 | 0.282 |
| QiD Law | 0.470 | 0.756 | 0.721 |
| Law of Precision | 0.294 | 0.744 | 0.713 |
| Symmetric Law | −0.445 | 0.712 | 0.717 |
| Asymmetric Law | −0.364 | 0.767 | 0.720 |

*Table 8.* Progressive Model Extrapolation on Pythia. We fit each law on the $k$ smallest models and predict the larger ones, reporting pooled $R^2$ across all 6 SNR levels.

**Model Extrapolation** We fit each law on the $k$ smallest Pythia models and predict the larger held-out models. Table 8 shows that at $k=4$ (training on [2.8B, 1B, 410M, 160M] to predict [12B, 6.9B]), Shannon-Simpl achieves pooled $R^2 = 0.837$ versus $0.756$ for the 9-parameter QiD Law. OpenAI's monotonic law collapses to $-0.048$, confirming that perturbation-agnostic forms cannot generalize to larger scales.

| Method | $k=4$, $j=8$ Predict [12B, 6.9B], 75.5B–307B | $k=5$, $j=12$ Predict 12B, 180.4B–307B | $k=5$, $j=15$ Predict 12B, 272.6B–307B |
|---|---|---|---|
| Shannon (Ours, 9p) | **0.590** | **0.847** | **0.828** |
| Shannon-Simpl (Ours, 6p) | 0.494 | 0.673 | 0.715 |
| Chinchilla | −0.146 | 0.305 | 0.439 |
| OpenAI | −0.341 | −0.082 | −0.034 |
| QiD Law | 0.550 | 0.761 | 0.650 |
| Law of Precision | 0.569 | 0.708 | 0.631 |
| Symmetric Law | 0.430 | 0.655 | 0.576 |
| Asymmetric Law | 0.525 | 0.611 | 0.558 |

*Table 9.* **Joint Model and Token Extrapolation.** We fit each law on the $k$ smallest models using only their first $j$ token checkpoints, then predict the held-out larger model(s) on larger token budgets. The 9-parameter Shannon Scaling Law dominates the 6-parameter Shannon-Simpl variant on joint extrapolation, confirming the extra parameters encode meaningful $(N, D)$ joint structure.

**Joint Extrapolation: Unseen Model and Unseen Tokens** The most stringent test extrapolates along both axes simultaneously. The headline configuration $k=5$, $j=12$ trains each law on the five smallest models using only their first 12 token checkpoints ($\leq$180B tokens), then predicts the held-out 12B model on 4 unseen token budgets spanning 180B–307B tokens. This requires generalizing to a model $1.7\times$ larger than anything in the training set, on a token range $1.7\times$ beyond the training horizon. As reported in Table 9, only the full Shannon Law maintains $R^2 = 0.847$ in this regime; Chinchilla degrades to $0.305$ and OpenAI to $-0.082$. Compared with Shannon-Simpl's $0.673$ on the same setting, this

confirms that the additional parameters in the full law do not merely overfit but encode meaningful joint $(N, D)$ structure that is identifiable only when training data spans both axes.

**When to Use Simplified vs Full Shannon Scaling Law**
The extrapolation results in subsection 5.2 suggest a clear practical guideline. Shannon-Simpl (6p) is preferred for *single-axis* extrapolation: with few held-out points to predict, its reduced parameterization avoids overfitting the limited variation. The full Shannon law (9p), in contrast, is necessary for *joint* $(N, D)$ extrapolation: its extra parameters separately control the signal scaling, the data–model interaction noise $c(DN)^\gamma$, and the irreducible noise floor, terms that become identifiable only when the training data spans both axes. The gap is decisive in the most stringent regime: 0.847 (Full) vs 0.673 (Simpl) at $k$=5, $j$=12. In typical practice, where one fits a $N \times D$ grid of small/short runs to predict performance at larger target $(N, D)$, the full Shannon law is the appropriate choice.

## 5.3. Fitted Exponents Reveal When Scaling Helps and When It Hurts

We investigate the fitted exponents in Table 5 to understand the competing dynamics between information gain and noise amplification.

**Under Noise, Model Scaling Backfires ($\gamma > \alpha$)**   The relationship between exponents of bandwidth ($\alpha$) and model noise ($\gamma$) exhibits a clear SNR-dependent crossover. In high-SNR regimes (e.g., Pretrain or GSM8K at 1e−5), $\alpha > \gamma$, confirming that capacity benefits outweigh noise. However, in low-SNR regimes, this inverts to $\gamma > \alpha$ (e.g., GSM8K at LR = 1e−4, where $0.475 > 0.321$). When perturbation gets larger, the model noise grows faster than the effective bandwidth. This empirical inversion aligns with the formation of the "loss basins" observed in the contour plots: beyond a certain threshold, scaling $N$ becomes detrimental as it amplifies the noise.

**Token Noise Always Wins in the Limit ($\delta > \beta$)**   Conversely, token dynamics reveal a persistent trend where the noise exponent $\delta$ consistently exceeds the signal exponent $\beta$ across all scenarios, including pretraining. This implies that the U-shaped degradation along the token axis is intrinsic rather than merely perturbation-induced: given a sufficiently large budget $D$, accumulated noise ($D^\delta$) will inevitably overshadow information gain ($D^\beta$), leading to performance degradation. Our exponent analysis indicates that U-shaped degradation exists universally.

## 5.4. Necessity of the $D$-$N$ Interaction Noise Term

The right panel of Table 6 validates the design of the model-interaction noise term $c(DN)^\gamma$ by isolating the impact of the token budget $D$. While the full interaction term and the size-only ablation ($cN^\gamma$) are indistinguishable in high-

SNR regimes (both achieving $R^2 \approx 0.99$ for Pythia at 40 dB), a sharp performance divergence emerges under heavy perturbation. Specifically, in the 10 dB scenario, the proposed $c(DN)^\gamma$ maintains robust alignment ($R^2 = 0.9555$), whereas the size-only baseline degrades significantly to 0.8035. Crucially, this coupled formulation allows our method to significantly outperform the Baseline Pareto Front, which drops to 0.8322 in the noise-dominated regime. These results empirically demonstrate that explicitly modeling the joint scaling dynamics of $D$ and $N$ is indispensable for capturing loss behaviors when noise dominates.

## 5.5. Incorporating the Perturbation Factor $X$

Perturbation-aware laws (Ouyang et al., 2022; Kumar et al., 2024) explicitly include $X$ as a variable. To demonstrate the structural integrity of our framework, we generalize our law by integrating $X$ directly into the SNR term:

$$C_{\text{ext}} = aN^\alpha \log_2 \left( 1 + \frac{X \cdot bD^\beta}{c(DN)^\gamma + dD^\delta + e} \right) \quad (13)$$

Quantitative analysis across 6 SNR levels (consistent with Table 2) confirms that this extension preserves robust modeling capabilities, achieving an average $R^2$ of $0.9602 \pm 0.03$. This statistically matches the original implicit formulation ($0.9613 \pm 0.03$) while significantly outperforming the best competing baseline ($0.9272 \pm 0.06$). Notably, the explicit inclusion of $X$ enhances alignment in high-noise regimes: improving the fit from 0.9234 to 0.9299 at 12 dB, while maintaining identical precision at high SNRs (0.9895 at 40 dB). This result validates that our proposed denominator term correctly models the intrinsic noise floor. Here, $X$ acts as an extrinsic SNR modulator, requiring no additional fitting parameters. This formulation provides flexibility for scenarios where external noise levels are known.

# 6. Conclusion

We propose the Shannon Scaling Law, a unified framework that reconceptualizes LLM training as information transmission, mapping parameters and tokens to channel bandwidth and signal power. Our analysis identifies a fundamental Shannon capacity, proving that scaling without maintaining sufficient SNR leads to the performance degradation observed in *catastrophic overtraining* and quantization, a phenomenon our model captures with superior fidelity compared to baselines. Crucially, the law also extrapolates $1.7\times$ beyond its fitting range: fitted on $\leq$6.9B Pythia models with $\leq$180B tokens, it predicts the unseen 12B model up to 307B tokens at pooled $R^2$=0.847, while monotonic baselines collapse. Ultimately, these findings advocate for a strategic shift from brute-force model expansion to prioritizing information density and SNR maximization.

## Impact Statement

This paper presents a unified scaling law for Large Language Models (LLMs) grounded in Shannon information theory. We view the work as primarily theoretical and methodological, advancing the field of machine learning. Below we briefly note potential broader impacts.

Positive societal impact. Training frontier LLMs consumes substantial energy and computational resources. Our framework can predict when additional model size or training tokens stop yielding gains and when they begin to degrade performance under perturbations such as quantization or fine-tuning. This helps practitioners avoid compute spent past the point of diminishing or negative return. In particular, our extrapolation results suggest that small-scale fitting runs can forecast the behaviour of much larger models, which may reduce wasteful pretraining experiments and the associated carbon footprint.

Risks and limitations. Similar to other scaling law research, our work could indirectly encourage the deployment of increasingly large LLMs, which have their own societal risks.

We do not foresee distinctive ethical concerns beyond those typical of foundational machine learning research, and we encourage practitioners to combine SNR-aware scaling analysis with established responsible AI practices.

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

# A. Appendix

## A.1. Implementation Details

| Model | Model Sizes $N$ | Pretrain Tokens $D$ |
|---|---|---|
| Pythia Suite | deduped-160m, deduped-410m, deduped-1b, deduped-2.8b, deduped-6.4b, deduped-12b | step: 2000, 6000, 10000, 14000, 20000, 24000, 29000, 36000, 43000, 57000, 71000, 86000, 98000, 120000, 130000, 140000 |
| OLMo2 | OLMo-2-0425-1B, stage1 | 10000-21, 40000-84, 100000-210, 240000-504, 480000-1007, 880000-1846, 1170000-2454, 1450000-3041, 1800000-3775, 1907359-4001 |
| | OLMo-2-1124-7B, stage1 | 4000-17, 19000-80, 55000-231, 120000-504, 240000-1007, 441000-1850, 480000-2014, 584000-2450, 727000-3050, 928646-3896 |
| | OLMo-2-1124-13B, stage1 | 3000-26, 10000-84, 60000-504, 143000-1200, 220000-1846, 292000-2450, 363000-3046, 450000-3775, 540000-4530, 596000-5000 |
| | OLMo-2-0325-32B, stage1 | 10000-84, 60000-504, 220000-1846, 292000-2450, 363000-3046, 450000-3775, 540000-4530, 596000-5000, 660000-5537, 720000-6040 |

*Table 10.* Summary of model configurations and pretraining steps.

| Task | Implementation | Parameters |
|---|---|---|
| SFT | `trl.SFTTrainer,`
`trl.SFTConfig` | `num_train_epochs=1,`
`per_device_train_batch_size=64,`
`per_device_eval_batch_size=64,`
`gradient_accumulation_steps=1,`
`optim="adamw_torch",`
`learning_rate=lr,`
`lr_scheduler_type="cosine",`
`save_strategy="steps",`
`eval_strategy="steps",`
`eval_steps=100,`
`bf16=args.bf16,`
`load_best_model_at_end=True,`
`max_length=1024,`
`warmup_ratio=0.1` |
| Quantization (GPTQ) | `optimum.gptq.GPTQQuantizer` | `bit=[2, 3, 4]` |
| Curve Fitting | `scipy.optimize.curve_fit` | `Initialization=[1] or [0.1]` |

*Table 11.* Implementation details.

## A.2. Gaussian noise over sizes

**Superior Fit Compared to the Best Baseline over** $N$    Figure 7 visualizes the fitting performance of our Shannon Scaling Law compared to the strongest baseline, the Quantization-induced Degradation (QiD) law. It illustrates the scaling dynamics across the entire spectrum of noise levels (from 40 dB down to 10 dB) for fixed token budgets. The plots reveal a transition from the "clean" regime (bottom rows), where scaling is monotonic, to the "noisy" regime (top rows), where the U-shape dominates. Our law consistently outperforms the QiD law, validating that the model noise term in our equation correctly scales with the intensity of perturbations. Under insufficient training regimes, where the model is trained with a limited

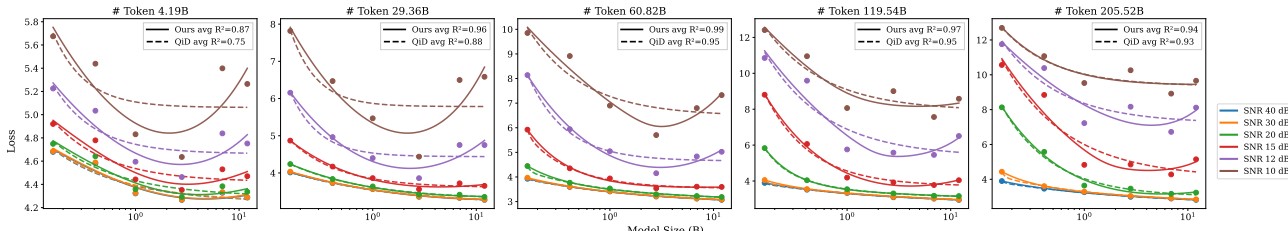

*Figure 7.* Fitting performance comparison: Shannon Law (Solid) vs. QiD Baseline (Dashed). The plots illustrate loss trajectories across varying Gaussian noise levels (10–40 dB) for fixed token budgets. While both laws align in high-SNR regimes (bottom curves), our Shannon Law demonstrates superior robustness in noise-dominated settings (top curves, e.g., 10 dB), tightly tracking the U-shaped degradation where the QiD baseline systematically deviates.

number of tokens, internal noise becomes more pronounced. In these cases, our law achieves average $R^2$ scores of $0.87$ and $0.95$, significantly surpassing QiD's $0.75$ and $0.88$. In each subgraph, our law tightly fits the data points across all noise levels simultaneously. Conversely, even though the QiD law is perturbation-aware, it still collapses in low SNR scenarios.

### A.3. 3D plots of Gaussian Noise Perturbations

**Visualizing Surface Alignment in Noise-Dominated Regimes.** Figure 8 presents a 3D visualization of the fitted loss surfaces, comparing our Shannon Scaling Law (orange surface) against the QiD baseline (blue wireframe). In high-SNR regimes (40 dB and 30 dB), both models exhibit high fidelity, with the blue wireframe and orange surface nearly overlapping, reflecting the effectiveness of both perturbation-aware law formulations. However, a critical divergence emerges as the SNR degrades. In the noise-dominated regimes of 12 dB and 10 dB, the empirical loss landscape exhibits a sharp, non-linear escalation at the boundaries (high $N$ and $D$), forming a steep "wall" of degradation.

Visual inspection reveals that the QiD law fails to accommodate this intense curvature. The blue wireframe becomes rigid and effectively detaches from the data manifold, significantly underestimating the loss values in the high-noise regions. In contrast, our Shannon formulation demonstrates superior geometric flexibility. The orange surface tightly adheres to the data points, accurately tracking the steep vertical ascent of the loss trajectory. This visual evidence aligns with the quantitative metrics, where our method maintains a high $R^2$ of $0.954$ at 10 dB, whereas the QiD law falters at $0.817$.

### A.4. Robustness across Quantization Schemes

To confirm that the Shannon Law's advantage is not specific to GPTQ, we additionally evaluate three off-the-shelf post-training quantization schemes on Pythia: AWQ (Lin et al., 2024), bitsandbytes (bnb) (Dettmers et al., 2023), and quanto [2]. AWQ and bnb are evaluated at 4-bit, and quanto is pushed to a more aggressive 2-bit regime. As reported in Table 12, the pattern observed under GPTQ is reproduced consistently: at AWQ and bnb 4-bit, all perturbation-aware laws perform well, but under 2-bit quanto the monotonic power laws collapse (OpenAI: $0.013$, Chinchilla: $0.028$), while the Shannon Law retains $R^2 = 0.9031$, the highest among all baselines, surpassing both QiD Law and Law of Precision. The choice of methods reflects compatibility with the Pythia architecture rather than cherry-picking.

| Method | AWQ 4-bit | bnb 4-bit | quanto 2-bit |
|---|---|---|---|
| Shannon (Ours) | **0.9935** | 0.9881 | **0.9031** |
| Shannon-Simpl (Ours) | 0.9837 | 0.9768 | 0.8454 |
| Chinchilla | 0.9574 | 0.9542 | 0.0276 |
| OpenAI | 0.9764 | 0.9679 | 0.0131 |
| QiD Law | 0.9855 | 0.9860 | 0.8931 |
| Law of Precision | 0.9855 | 0.9860 | 0.8932 |
| Symmetric Law | 0.9866 | 0.9867 | 0.8531 |
| Asymmetric Law | 0.9913 | **0.9936** | 0.8636 |

*Table 12.* Robustness across post-training quantization schemes on Pythia (AWQ, bitsandbytes, quanto). Under aggressive 2-bit quanto, monotonic power laws collapse ($R^2 \approx 0.01$–$0.03$), while the Shannon Law preserves $R^2 = 0.9031$, the highest among all baselines.

---

[2] https://github.com/huggingface/optimum-quanto

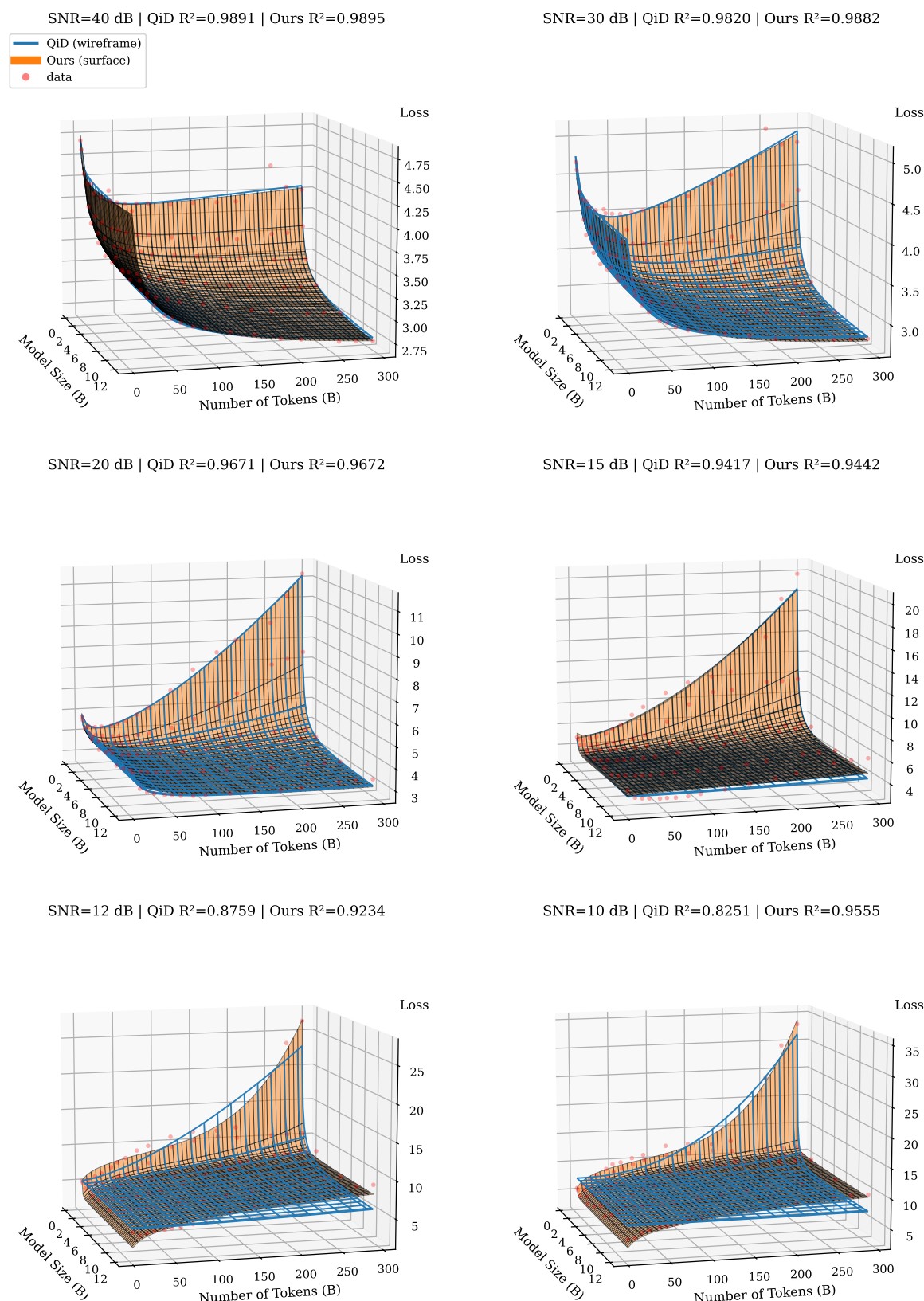

*Figure 8.* 3D Loss Landscapes: ours (orange surface) vs. QiD law (blue wireframe). Our method demonstrates superior geometric flexibility, maintaining high fidelity ($R^2 = 0.9555$) even in the extreme 10 dB regime where the baseline ($R^2 = 0.8251$) fails to track the steep curvature of the data manifold.

**A.5. SFT as Perturbation: Contour plots of all learning rates**

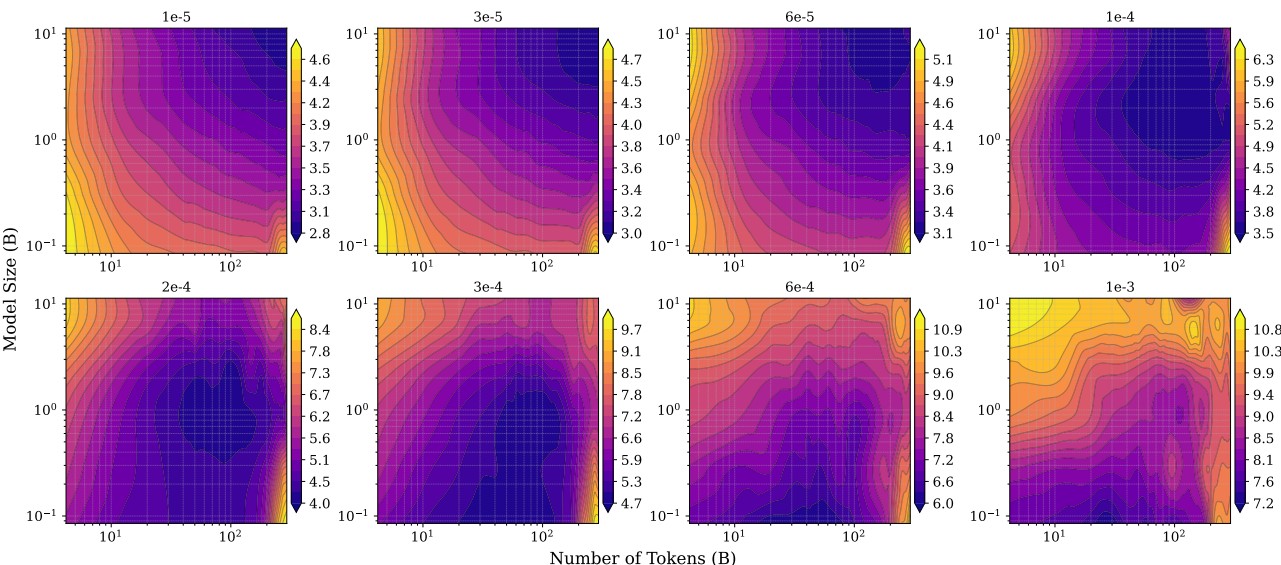

*Figure 9.* Evolution of Pythia loss contours on GSM8K under SFT with varying learning rates (8 values). As the learning rate increases, corresponding to stronger optimization noise, the loss landscape transitions from a smooth monotonic regime to a noise-dominated basin. In this regime, increasing model size ($N$) or training tokens ($D$) fails to improve performance, illustrating the breakdown of conventional scaling under extreme perturbations.

**Emergence of U-shaped Curves and Loss Basins under SFT Perturbation.**    Figure 9, Figure 10 and Figure 11 visualizes the evolution of loss landscapes for GSM8K, SiQA, and StarCoder as the learning rate (LR) increases from $1e-5$ to $1e-3$. Each set of 8 contour plots reveals a fundamental transition in scaling dynamics under SFT perturbation.

In the low-LR regime, the landscapes exhibit traditional monotonic scaling, where increasing model size ($N$) or token budget ($D$) consistently reduces loss. However, as the LR intensifies, the monotonicity breaks, and a distinct "loss basin" emerges in the center (e.g. at LR $= 2e-4$ of GSM8K, LR $= 1e-4$ of SiQA and at LR $= 3e-4$ of StarCoder). This basin represents a closed region of optimal performance surrounded by degradation, confirming the emergence of U-shaped curves along both axes:

- **Along the Model Size ($N$):** For a fixed token budget, scaling up the model size initially reduces loss but eventually leads to degradation. It is visible as the color shifting back to yellow at the top of the plots. This validates our hypothesis that excessively large models contain more model noise.

- **Along the Token Number ($D$):** Similarly, for a fixed model size, increasing training tokens beyond the optimal point incurs a penalty, likely due to the accumulation of noise during extended training steps.

At the highest LR, the system undergoes catastrophic collapse. The low-loss basin vanishes, replaced by high loss values across both $N$ and $D$ dimensions. This universal pattern is observed consistently across GSM8K, SiQA, and StarCoder. It strongly validates our Shannon Scaling Law: the noise term in the denominator eventually dominates the capacity term, no matter with larger $N$ or large $D$.

**A.6. SFT as Perturbation: GSM8K, SiQA and StarCoder.**

We further evaluate the generalize ability of our law on SiQA (Commonsense QA) and StarCoder (Code Generation), as detailed in Table 14 and Table 15. Consistent with the GSM8K results, our Shannon Scaling Law achieves the highest goodness-of-fit across both datasets, yielding average $R^2$ scores of $0.9252$ on SiQA and $0.9055$ on StarCoder. In contrast, the strongest competing baseline (Asymmetric Law, Row 7) lags behind with averages of $0.8948$ and $0.8841$, respectively. The robustness of our formulation is particularly evident in challenging regimes. For instance, on SiQA at LR $= 1e-4$, our law maintains a robust fit of $0.8305$, significantly outperforming the best baseline's $0.7583$. Furthermore, traditional

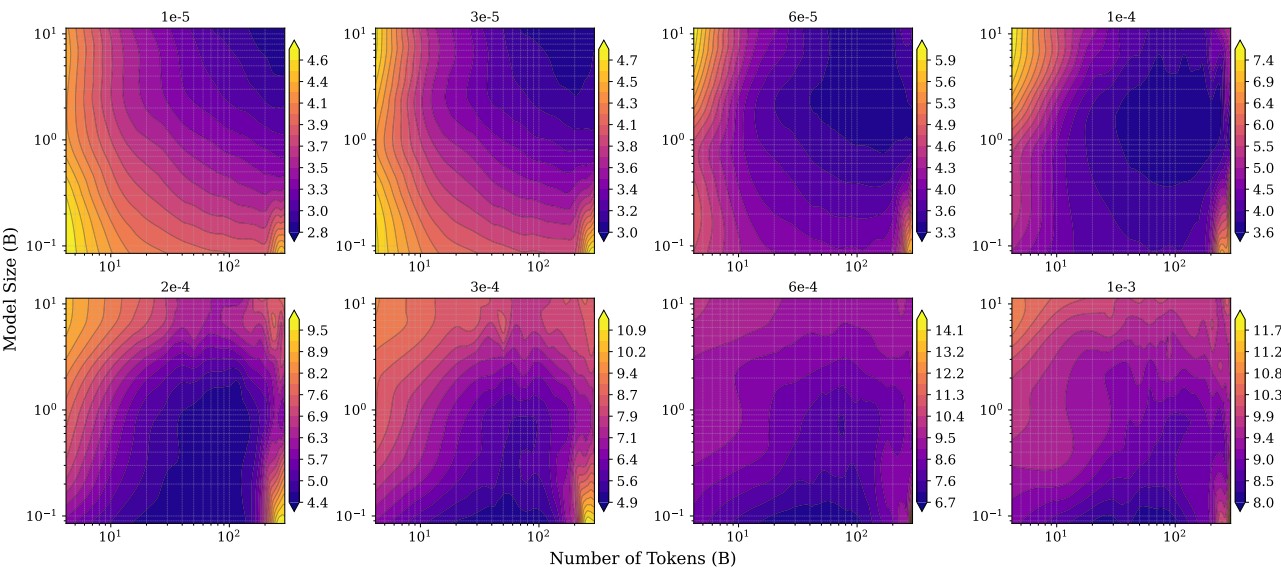

*Figure 10.* Evolution of Pythia loss contours on SiQA under SFT with varying learning rates (8 values). As the learning rate increases, corresponding to stronger optimization noise, the loss landscape transitions from a smooth monotonic regime to a noise-dominated basin. In this regime, increasing model size ($N$) or training tokens ($D$) fails to improve performance, illustrating the breakdown of conventional scaling under extreme perturbations.

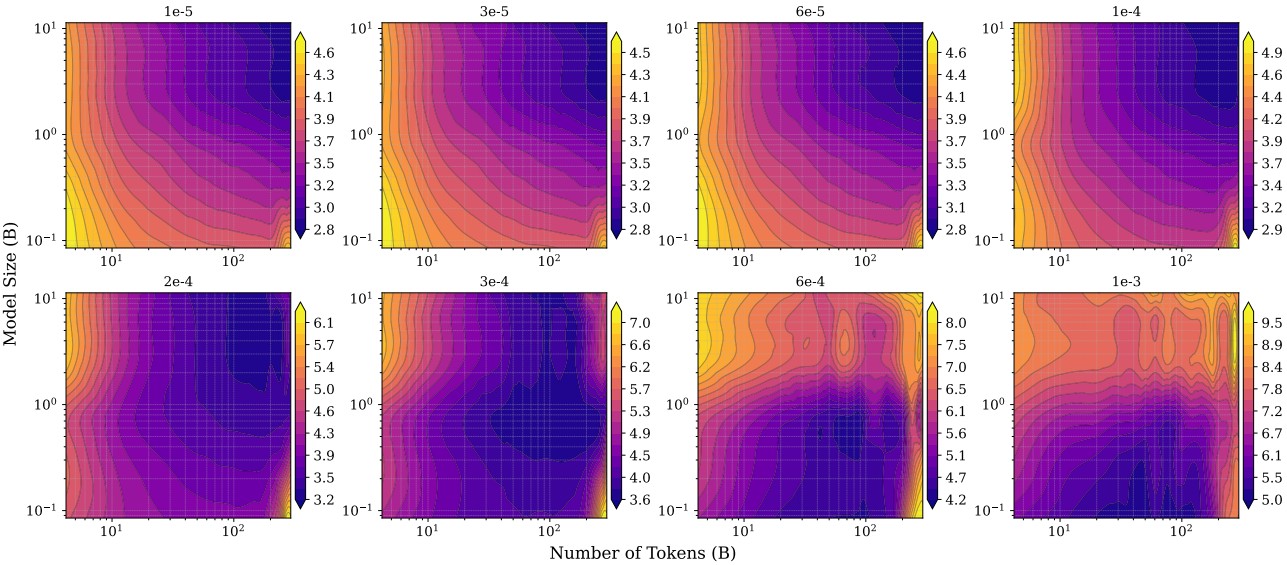

*Figure 11.* Evolution of Pythia loss contours on StarCoder under SFT with varying learning rates (8 values). As the learning rate increases, corresponding to stronger optimization noise, the loss landscape transitions from a smooth monotonic regime to a noise-dominated basin. In this regime, increasing model size ($N$) or training tokens ($D$) fails to improve performance, illustrating the breakdown of conventional scaling under extreme perturbations.

| GSM8K | $R^2$ | | | | | | | | |
|---|---|---|---|---|---|---|---|---|---|
| L(N, D) | LR=1e-5 | LR=3e-5 | LR=6e-5 | LR=1e-4 | LR=2e-4 | LR=3e-4 | LR=6e-4 | LR=1e-3 | Avg ± Std |
| $\frac{1}{aN^\alpha \log_2\left(1+\frac{bD^\beta}{c(DN)^\gamma+dD^\delta+e}\right)}$ | **0.9856** | 0.9809 | 0.9704 | **0.9010** | **0.8736** | **0.9494** | **0.9504** | **0.8778** | **0.9361 ± 0.05** |
| $\left[\left(\frac{a}{N}\right)^{\frac{\alpha}{\beta}}+\frac{b}{D}\right]^\beta$ | 0.9625 | 0.9412 | 0.8472 | 0.4660 | 0.0000 | -8.9821 | -0.5793 | -0.8414 | -0.8982 ± 3.34 |
| $\frac{a}{N^\alpha}+\frac{b}{D^\beta}+c$ | 0.9363 | 0.9110 | 0.8134 | 0.4771 | 0.0195 | 0.0054 | 0.0000 | 0.0057 | 0.3961 ± 0.44 |
| $\frac{a}{N^\alpha}+\frac{b}{D^\beta}+c+d\frac{D^{\beta'}}{N^{\alpha'}}$ | 0.7976 | **0.9857** | **0.9748** | 0.8855 | 0.3975 | 0.3649 | 0.2366 | 0.1005 | 0.5929 ± 0.36 |
| $a\frac{N^\alpha}{D^\beta}+b\frac{N^\beta}{D^\alpha}+c$ | 0.9702 | 0.9536 | 0.8922 | 0.6843 | 0.6590 | 0.7105 | 0.8020 | 0.7716 | 0.8054 ± 0.12 |
| $a\frac{N^\alpha}{D^\beta}+b\frac{D^{\beta'}}{N^{\alpha'}}+c$ | 0.9779 | 0.9656 | 0.9332 | 0.8191 | 0.8036 | 0.9249 | 0.9438 | **0.8778** | 0.9057 ± 0.07 |

*Table 13.* Goodness-of-fit ($R^2$) comparison on GSM8K under varying SFT learning rates (8 values). The Shannon Law remains robust across the U-shaped loss landscape.

| SiQA | Pythia | | | | | | | | |
|---|---|---|---|---|---|---|---|---|---|
| L(N, D) / L(N, D, X) | LR=1e-5 | LR=3e-5 | LR=6e-5 | LR=1e-4 | LR=2e-4 | LR=3e-4 | LR=6e-4 | LR=1e-3 | Avg ± Std |
| $\frac{1}{aN^\alpha \log_2\left(1+\frac{bD^\beta}{c(DN)^\gamma+dD^\delta+e}\right)}$ | **0.9865** | **0.9784** | **0.9240** | **0.8305** | **0.9198** | **0.9300** | **0.9205** | **0.9120** | **0.9252 ± 0.05** |
| $\left[\left(\frac{a}{N}\right)^{\frac{\alpha}{\beta}}+\frac{b}{D}\right]^\beta$ | 0.9562 | 0.9059 | 0.5799 | 0.0000 | 0.0000 | -0.4779 | -0.8789 | -1.9634 | -0.1098 ± 0.99 |
| $\frac{a}{N^\alpha}+\frac{b}{D^\beta}+c$ | 0.9283 | 0.8832 | 0.5795 | 0.1667 | 0.0136 | 0.0061 | 0.0072 | 0.0122 | 0.3246 ± 0.41 |
| $\frac{a}{N^\alpha}+\frac{b}{D^\beta}+c+d\frac{D^{\beta'}X^\gamma}{N^{\alpha'}}$ | 0.9294 | 0.8723 | 0.6484 | 0.3791 | 0.4877 | 0.3717 | 0.4270 | 0.2896 | 0.5507 ± 0.24 |
| $\frac{a}{N^\alpha}+\frac{b}{D^\beta}+c+d\frac{D^{\beta'}e^{\gamma X}}{N^{\alpha'}}$ | 0.9853 | 0.9670 | 0.8679 | 0.7476 | 0.4877 | 0.3655 | 0.2226 | 0.2757 | 0.6149 ± 0.31 |
| $a\frac{N^\alpha}{D^\beta}+b\frac{D^\beta}{N^\alpha}+c$ | 0.9645 | 0.9278 | 0.7347 | 0.6634 | 0.6827 | 0.6530 | 0.7569 | 0.7477 | 0.7663 ± 0.12 |
| $a\frac{N^\alpha}{D^\beta}+b\frac{D^{\beta'}}{N^{\alpha'}}+c$ | 0.9743 | 0.9503 | 0.8382 | 0.7583 | 0.8856 | 0.9218 | 0.9203 | 0.9094 | 0.8948 ± 0.07 |

*Table 14.* Goodness-of-fit ($R^2$) comparison on SiQA under varying SFT learning rates (8 values). The Shannon Law remains robust across the U-shaped loss landscape.

| StarCoder | Pythia | | | | | | | | |
|---|---|---|---|---|---|---|---|---|---|
| L(N, D) / L(N, D, X) | LR=1e-5 | LR=3e-5 | LR=6e-5 | LR=1e-4 | LR=2e-4 | LR=3e-4 | LR=6e-4 | LR=1e-3 | Avg ± Std |
| $\frac{1}{aN^\alpha \log_2\left(1+\frac{bD^\beta}{c(DN)^\gamma+dD^\delta+e}\right)}$ | 0.9769 | 0.9758 | **0.9757** | **0.9728** | 0.9010 | 0.6848 | **0.8577** | **0.8993** | **0.9055 ± 0.10** |
| $\left[\left(\frac{a}{N}\right)^{\frac{\alpha}{\beta}}+\frac{b}{D}\right]^\beta$ | 0.9564 | 0.9525 | 0.9417 | 0.8806 | 0.5070 | -0.0729 | -8.3358 | -0.5998 | -0.5963 ± 3.18 |
| $\frac{a}{N^\alpha}+\frac{b}{D^\beta}+c$ | 0.9283 | 0.9191 | 0.9026 | 0.8318 | 0.5071 | 0.1131 | 0.0020 | 0.0000 | 0.5255 ± 0.43 |
| $\frac{a}{N^\alpha}+\frac{b}{D^\beta}+c+d\frac{D^{\beta'}X^\gamma}{N^{\alpha'}}$ | 0.9287 | 0.9196 | 0.8987 | 0.8443 | 0.6046 | **0.6961** | 0.3842 | 0.0072 | 0.6604 ± 0.32 |
| $\frac{a}{N^\alpha}+\frac{b}{D^\beta}+c+d\frac{D^{\beta'}e^{\gamma X}}{N^{\alpha'}}$ | **0.9796** | **0.9778** | 0.9739 | 0.9610 | **0.9131** | 0.6838 | 0.3842 | 0.1945 | 0.7585 ± 0.31 |
| $a\frac{N^\alpha}{D^\beta}+b\frac{D^\beta}{N^\alpha}+c$ | 0.9628 | 0.9587 | 0.9498 | 0.9042 | 0.6766 | 0.5026 | 0.6413 | 0.7976 | 0.7992 ± 0.17 |
| $a\frac{N^\alpha}{D^\beta}+b\frac{D^{\beta'}}{N^{\alpha'}}+c$ | 0.9698 | 0.9670 | 0.9631 | 0.9408 | 0.8722 | 0.6313 | 0.8317 | 0.8971 | 0.8841 ± 0.11 |

*Table 15.* Goodness-of-fit ($R^2$) comparison on StarCoder under varying SFT learning rates (8 values). The Shannon Law remains robust across the U-shaped loss landscape.

power laws (Rows 2–3) exhibit catastrophic failure on these tasks, frequently yielding negative $R^2$ values (e.g., an average of $-0.1098$ and $-0.5998$ for OpenAI law on SiQA and StarCoder respectively), which empirically validates that these laws are incapable of modeling the complex dynamics under SFT which plays a role of perturbation.

