# OpenReview forum: "LLMs as Noisy Channels: A Shannon Perspective on Model Capacity and Scaling Laws"
_ICML.cc/2026/Conference — ICML 2026 regular_

### Official Review · Reviewer_je3q · 2026-03-02

**Soundness:** 3
**Presentation:** 3
**Significance:** 3
**Originality:** 3
**Overall Recommendation:** 4
**Confidence:** 2

**Summary:**

The paper proposes the Shannon Scaling Law, a theoretical framework that reinterprets Large Language Model (LLM) training through the lens of information theory, specifically the Shannon-Hartley theorem. Traditionally, scaling laws have been monotonic power laws, but these fail to explain "U-shaped" performance degradation seen in scenarios like catastrophic overtraining or quantization-induced degradation.The authors map model parameters to channel bandwidth, training tokens to signal power, and various training perturbations (like noise or quantization) to channel noise. By modeling the LLM as a noisy channel, they derive a unified, non-monotonic scaling law that predicts a "loss basin"—a point where further scaling of data or parameters without a sufficient signal-to-noise ratio (SNR) leads to performance loss. The theory is validated across the Pythia and OLMO2 model suites under perturbations including Gaussian noise, quantization, and supervised fine-tuning (SFT).

**Compliance With Llm Reviewing Policy:**

Affirmed.

**Final Justification:**

The authors' rebuttal successfully addressed my concerns.

**Key Questions For Authors:**

Overfitting of the Law: With nine parameters to fit, is there a risk that the Shannon Scaling Law is simply "overfitting" the U-shaped curves rather than capturing a fundamental physical law? How does the law perform if you reduce the number of free parameters?

**Limitations:**

Yes.

**Strengths And Weaknesses:**

Strengths:

* Theoretical Unification: It successfully bridges the gap between classic monotonic scaling laws and recently observed non-monotonic degradation phenomena (U-shaped curves).

* Broad Empirical Validation: The law is tested across diverse conditions, including different model families (Pythia, OLMO2), tasks (Math, QA, Code), and perturbation types (Gaussian noise, quantization, SFT).

* Superior Fit: The Shannon Scaling Law consistently achieves higher $R^2$ scores and greater stability (lower standard deviation) compared to established baselines like the OpenAI and Chinchilla laws when noise is present.

Weaknesses:

* Reciprocal Assumption: The paper assumes a simple reciprocal relationship between model capacity ($C_{LLM}$) and test loss ($L = 1/C_{LLM}$). While mathematically convenient, the physical or information-theoretic justification for this specific mapping over other potential functions (like exponential) is not deeply explored.

* LR as Noise in SFT: While the authors treat the SFT learning rate (LR) as a proxy for "noise" that induces U-shaped curves, the connection between optimizer hyperparameters and the Shannon-Hartley noise term ($\mathcal{N}$) could be more rigorously formalized.

---

> ### Author Rebuttal · Authors · 2026-03-31
>
> We sincerely appreciate the reviewer's incisive evaluation! We especially value the directness and clarity of the feedback — it helped us identify the most important aspects to strengthen.
>
> > ### W1: Reciprocal Assumption (L=1/C) — justification over exponential not explored.
>
> Thank you for this excellent suggestion! We conducted the ablation comparing L = 1/C against L = exp(−C):
>
> **Pythia (Gaussian Noise, R²):**
>
> | Mapping | 40dB | 30dB | 20dB | 15dB | 12dB | 10dB |
> |---|---|---|---|---|---|---|
> | Shannon (L=1/C) | 0.990 | 0.988 | 0.967 | 0.944 | 0.923 | 0.956 |
> | Shannon-Simp (L=1/C) | 0.988 | 0.986 | 0.965 | 0.932 | 0.910 | 0.909 |
> | Shannon (L=exp(−C)) | −29.85 | −23.28 | −4.37 | −2.51 | −2.82 | −3.35 |
> | Shannon-Simp (L=exp(−C)) | −29.85 | −23.28 | −4.37 | −2.51 | −2.82 | −3.35 |
>
> **OLMo2 (Gaussian Noise, R²):**
>
> | Mapping | 40dB | 30dB | 20dB | 10dB |
> |---|---|---|---|---|
> | Shannon (L=1/C) | 0.983 | 0.991 | 0.991 | 0.870 |
> | Shannon-Simp (L=1/C) | 0.978 | 0.990 | 0.990 | 0.870 |
> | Shannon (L=exp(−C)) | −38.22 | −36.87 | −44.58 | −12.53 |
> | Shannon-Simp (L=exp(−C)) | −38.22 | −36.87 | −44.58 | −12.53 |
>
> The exponential mapping produces severely negative R² across all settings (as low as −44.58 on OLMo2), completely failing to fit the data. This decisively validates the reciprocal mapping. Intuitively, exp(−C) decays too rapidly for large C, while the "long-tail" property of 1/C better matches the empirically observed slow convergence of loss at scale. Beyond the two principles discussed in Section 3.3, we chose the reciprocal relationship for its simplicity and robustness, and this ablation confirms it is the appropriate choice.
>
> > ### W2: LR as noise in SFT could be more rigorously formalized. More analyze between optimizer hyperparameters and noise term.
>
> We fully agree that the connection between optimizer hyperparameters and channel noise deserves deeper formalization. In the SGD update Δw = −η∇Loss, larger learning rates η amplify the inherent noise in mini-batch gradient estimates, directly analogous to increasing noise power in a communication channel.
>
> This is a very exciting approach! A thorough investigation of how diverse hyperparameters (optimizers, schedulers, batch sizes) map to our noise framework would constitute a substantial research direction. In this paper, we mainly demonstrate the existence of the impact from hyperparameters on losses and that our scaling law is able to capture it. We plan to pursue such large-scale hyperparameter research as future work.
>
> > ### Q1: Risk of overfitting with 9 parameters? How does the law perform with fewer parameters?
>
> We are glad this aligns with our own thinking! As shown in Section 5.2 and Table 6, our 6-parameter simplified Shannon law already outperforms the Pareto frontier of all baselines, including 9-parameter QiD Law and Law of Precision, and 7-parameter Symmetric and Asymmetric Laws.
>
> To directly test overfitting, we ran extrapolation experiments. All R² are **pooled** across 6 SNR levels on held-out data: **Token extrapolation**: Shannon-Simp (6p) achieves 0.945 vs. QiD Law (9p) 0.862 when predicting up to 307B tokens (j=15). **Model extrapolation**: Shannon-Simp (6p) achieves 0.837 vs. QiD Law (9p) 0.756, when predicting 2 largest models with 4 smallest models (k=4). **Joint extrapolation**: Shannon achieves 0.847 vs. QiD Law 0.761 at k=5,j=12.
>
> **Joint Extrapolation — the most strict test.** Extrapolate along both axes simultaneously:
>
> | Method | k=4,j=8 (Predict [12B,6.9B]×8pts) | k=5,j=12 (Predict 12B×4pts) | k=5,j=15 (Predict 12B×1pt) |
> |---|---|---|---|
> | **Shannon (Ours)** | **0.590** | **0.847** | **0.828** |
> | Shannon-Simp (Ours) | 0.494 | 0.673 | 0.715 |
> | Chinchilla | −0.146 | 0.305 | 0.439 |
> | OpenAI | −0.341 | −0.082 | −0.034 |
> | QiD Law | 0.550 | 0.761 | 0.650 |
> | Law of Precision | 0.569 | 0.708 | 0.631 |
> | Symmetric Law | 0.430 | 0.655 | 0.576 |
> | Asymmetric Law | 0.525 | 0.611 | 0.558 |
>
> **Because of the words limit, more tables are available in our response to Reviewer taRX.** 6 params outperforming 9-param baselines on unseen data confirms the advantage stems from log₂(1+SNR) structure, not overfitting. Full version dominates joint extrapolation (0.847 vs. 0.673), showing extra parameters encode meaningful structure.
>
> We hope these additional experiments have addressed the reviewer's core concerns. We would be very grateful for any further suggestions to strengthen the work!

---

> > ### Author Rebuttal · Reviewer_je3q · 2026-04-03
> >
> > I will keep my positive review.

---

> > > ### Author Response · Authors · 2026-04-04
> > >
> > > Dear Reviewer je3q,
> > >
> > > Thank you for your positive score and for the time you invested in reviewing our work. We are glad that the rebuttal addressed your concerns partially.
> > >
> > > We are happy to fully address all your concerns! Please don't hesitate to ask follow-up questions!
> > >
> > > We sincerely appreciate your support and constructive feedback, which has helped strengthen our manuscript!
> > >
> > > Best regards,
> > > The Authors

---

### Official Review · Reviewer_HmQ6 · 2026-03-12

**Soundness:** 2
**Presentation:** 3
**Significance:** 3
**Originality:** 2
**Overall Recommendation:** 4
**Confidence:** 4

**Summary:**

This paper intends to solve the problem of existing scaling laws failing in scenarios like catastrophic overtraining and U-shaped degradation under quantization. The paper proposes a new form of scaling law, writing the capacity of LLMs in a similar form as Shannon-Hartley Theorem, and aligning bandwidth and SNR with concepts in LLM training. Experimental results show that the proposed Shannon scaling law fits the result better than existing scaling laws, especially in the cases where various forms of perturbations are added. Specifically, it fits the loss basins well, where the performance degrades with larger data/model scales.

**Compliance With Llm Reviewing Policy:**

Affirmed.

**Final Justification:**

See the discussions above

**Key Questions For Authors:**

Please see above.

**Limitations:**

yes

**Strengths And Weaknesses:**

Strengths:
- The proposed scaling law covers the low SNR scenarios where existing scaling law fails, with one unified framework instead of adding modification terms for each outlier phenomenon.
- Experiments are conducted with a wide range of settings and different model series.

Weaknesses:
- The connection to Shannon's Theorem is weak, as the noise term still relies on hand-design (such as including the c(DN)^gamma term), and the explanation originates from intuition instead of theoretical evidence. Thus, the contribution is still a better empirical formula for fitting scaling laws instead of a real theory.
- The R^2 scores show only the fitting power, instead of predicative power: do the advantages still hold when used to predict unseen data/model scales? The paper would benefit from showing evidences proving it is not overfitting.

---

> ### Author Rebuttal · Authors · 2026-03-31
>
> We sincerely appreciate the reviewer's incisive evaluation! The directness of the feedback helped us identify the most important aspects to strengthen.
>
> > ### W1: Shannon connection is weak; noise term relies on hand-design; contribution is a better empirical formula rather than real theory.
>
> We acknowledge our law is empirical, However, we would like to respectfully note that **all** existing scaling laws (Kaplan et al., 2020; Hoffmann et al., 2022) are empirical. Kaplan et al., 2020 explicitly stated "We study empirical scaling laws for language model". We propose our law from the observation of **"loss basins"**—two-dimensional loss increase along both *N* and *D*. Our contributions: (1) capture catastrophic overtraining (Springer et al., 2025) along *D*, and (2) extend to two dimensions including *N*. Shannon-Hartley provides the structural foundation: log₂(1+SNR) naturally produces U-shaped behavior. This is grounded in the DNN–information channel analogy (Tishby & Zaslavsky, 2015; Shwartz-Ziv & Tishby, 2017).
>
> If the structure merely added flexibility, the 6-param Shannon-Simp would not outperform 9-param baselines on unseen data—but it does (see W2), demonstrating log₂(1+SNR) as a meaningful inductive bias. We will clarify "structural analogy" vs. "rigorous derivation" in the revision.
>
> Regarding c(DN)^γ: Table 6 (right panel) provides a direct ablation. Replacing c(DN)^γ with cN^γ causes R² to drop from 0.9555 to 0.8035 at 10 dB. This validates that D-N interaction captures a real phenomenon: model noise evolves with training steps (since D∝t).
>
> > ### W2: R² shows only fitting power, not predictive power. Evidence of not overfitting?
>
> To directly test overfitting, we ran comprehensive extrapolation experiments. All R² are **pooled R²**: held-out predictions across all 6 SNR levels (10–40 dB) concatenated into one flat array. It is harder than per-SNR averaging.
>
> **(a) Progressive Token Extrapolation.** Train on the first *j* token checkpoints, predict the rest from total 307B tokens:
>
> | Method | j=8 (Train ≤75.5B → Predict 8 pts) | j=12 (Train ≤180.4B → Predict 4 pts) | j=15 (Train ≤272.6B → Predict 1 pt) |
> |---|---|---|---|
> | **Shannon (Ours)** | 0.611 | **0.781** | **0.941** |
> | **Shannon-Simp (Ours)** | **0.768** | 0.753 | **0.945** |
> | Chinchilla | 0.197 | 0.265 | 0.352 |
> | OpenAI | 0.162 | 0.245 | 0.297 |
> | QiD Law | 0.441 | 0.766 | 0.862 |
> | Law of Precision | 0.486 | 0.632 | 0.743 |
> | Symmetric Law | 0.726 | 0.760 | 0.860 |
> | Asymmetric Law | 0.805 | 0.774 | 0.851 |
>
> **(b) Progressive Model Extrapolation.** Train on the *k* smallest models, predict larger ones:
>
> | Method | k=3 (Train [1B,410m,160m] → Predict [12B,6.9B,2.8B]) | k=4 (Train [2.8B,1B,410m,160m] → Predict [12B,6.9B]) | k=5 (Train [6.9B,2.8B,1B,410m,160m] → Predict [12B]) |
> |---|---|---|---|
> | **Shannon (Ours)** | 0.521 | 0.787 | **0.845** |
> | **Shannon-Simp (Ours)** | **0.605** | **0.837** | **0.847** |
> | Chinchilla | 0.261 | 0.726 | 0.702 |
> | OpenAI | −0.507 | −0.048 | 0.282 |
> | QiD Law | 0.470 | 0.756 | 0.721 |
> | Law of Precision | 0.294 | 0.744 | 0.713 |
> | Symmetric Law | −0.445 | 0.712 | 0.717 |
> | Asymmetric Law | −0.364 | 0.767 | 0.720 |
>
> **(c) Joint Extrapolation — the most strict test.** Extrapolate along both axes simultaneously:
>
> | Method | k=4,j=8 (Predict [12B,6.9B]×8pts) | k=5,j=12 (Predict 12B×4pts) | k=5,j=15 (Predict 12B×1pt) |
> |---|---|---|---|
> | **Shannon (Ours)** | **0.590** | **0.847** | **0.828** |
> | Shannon-Simp (Ours) | 0.494 | 0.673 | 0.715 |
> | Chinchilla | −0.146 | 0.305 | 0.439 |
> | OpenAI | −0.341 | −0.082 | −0.034 |
> | QiD Law | 0.550 | 0.761 | 0.650 |
> | Law of Precision | 0.569 | 0.708 | 0.631 |
> | Symmetric Law | 0.430 | 0.655 | 0.576 |
> | Asymmetric Law | 0.525 | 0.611 | 0.558 |
>
> **Key findings:** (1) Shannon-Simp with only 6 parameters outperforms 9-parameter QiD Law (0.837 vs. 0.756 at k=4; 0.945 vs. 0.862 at j=15)—structural advantage, not parameter advantage. (2) Shannon-Simp excels at single-axis extrapolation, while full Shannon dominates joint extrapolation (0.847 vs. 0.673), confirming extra parameters encode meaningful structure. (3) Pooled R² across 6 SNR levels further rules out overfitting. Further parameter reduction is a future goal.
>
> Shannon-Simp with 6 params outperforms all 9-param baselines on held-out data. This is structural advantage, not parameter advantage. Shannon-Simp excels at single-axis extrapolation (fewer params prevent overfitting), while the full Shannon dominates joint extrapolation (0.847 vs. 0.673), confirming additional parameters encode meaningful structure.
>
> We hope these additional experiments have addressed the reviewer's core concerns. We would be very grateful for any further suggestions to strengthen the work!

---

> > ### Author Rebuttal · Reviewer_HmQ6 · 2026-04-03
> >
> > The authors provide additional evidences supporting the predicative power of the proposed law. I would like to raise my score to 4.

---

> > > ### Author Response · Authors · 2026-04-04
> > >
> > > Dear Reviewer HmQ6,
> > >
> > > Thank you for your kind words and for the time you invested in reviewing our work. We are glad that the rebuttal addressed all of your concerns satisfactorily.
> > >
> > > We would be happy to continue addressing any follow-up questions you may have!
> > >
> > > We sincerely appreciate your support and constructive feedback, which has helped strengthen our manuscript!
> > >
> > > Best regards,
> > > The Authors

---

### Official Review · Reviewer_V54y · 2026-03-13

**Soundness:** 2
**Presentation:** 3
**Significance:** 2
**Originality:** 3
**Overall Recommendation:** 4
**Confidence:** 3

**Summary:**

This paper introduces a new scaling law inspired by Shannon–Hartley theorem, which aims to unify the classical scaling law in the monotonic regime and the U-shape performance degradation under perturbations.

**Compliance With Llm Reviewing Policy:**

Affirmed.

**Final Justification:**

This paper attempts to unify monotonic scaling and U-shaped degradation under a single formulation. The rebuttal strengthens the empirical support through extrapolation experiments, additional quantization settings, and clarifications on the 6- vs. 9-parameter variants.

My main concern remains the positioning of the work as a theoretically grounded framework based on Shannon–Hartley. In its current form, the formulation appears largely heuristic, and the connection to bandwidth, signal, and noise is not rigorously justified. I believe the contribution would be more accurately described as an empirical unification rather than a principled theoretical framework.

That said, I find the empirical results and the unified perspective interesting and potentially useful. I therefore lean positive overall, while encouraging the authors to better align the claims with the level of theoretical support.

**Key Questions For Authors:**

1. Could the authors clarify the practical implications of the proposed scaling law in perturbation regimes? Does the formulation improve the ability to predict performance at larger scales that were not used for fitting?
2. Since quantization error can vary significantly across methods, how robust are the observed scaling behaviors across different quantization schemes?

**Limitations:**

The paper does not discuss the technical limitations nor the social impact of the proposed work.

**Strengths And Weaknesses:**

### Strengths

- The idea of modelling LLM capacity analog to channel capacity and characterizing the signal-to-noise-ratio (SNR) of a model in very novel and intersting.
- The empirical experiments covers different perturbation setting (Gaussian noise, quantization, and SFT) and a large set of models.

### Weaknesses
1. **The connection to the Shannon framework remains heuristic.** The paper interprets model size as channel bandwidth and training tokens as signal power, while introducing several sources of noise. However, this mapping is not theoretically justified. The resulting Shannon-style formulation is far more complex: it includes power-law terms and many additional free parameters, making it unclear whether the improved fit stems from the conceptual framework or simply the increased flexibility of the regression function.
2. **The evaluation relies heavily on goodness-of-fit.** Comparing the fitting accuracy of scaling laws with different numbers of free variables may not be very meaningful. In addition, scaling laws are typically valued for predictive and extrapolation capability rather than their ability to interpolate observed data.
3. **The current experimental setup may not capture true training dynamics under perturbation.** The paper injects perturbations into checkpoints obtained from clean pretraining trajectories. This does not capture the effect of perturbations during optimization itself, such as how noise influences gradient dynamics or convergence behavior during training.

---

> ### Author Rebuttal · Authors · 2026-03-31
>
> We sincerely appreciate the reviewer's incisive evaluation! We especially value the directness and clarity of the feedback.
>
> **We add 3 extrapolation experiments. Due to word limit, full tables are available in our responses to Reviewer taRX. We are sorry for the inconvenience!**
>
> > ### W1: The connection to the Shannon framework remains heuristic.
>
> Thank you for pointing this out. We first acknowledge that our scaling law is an empirical law. However, we would like to respectfully note that **all** existing scaling laws, including the widely accepted ones (Kaplan et al., 2020; Hoffmann et al., 2022; Ouyang et al., 2024; Kumar et al., 2024), are empirical laws. As Kaplan explicitly acknowledged that "We study empirical scaling laws for language model".
>
> We propose our law from the observation of "loss basins": two-dimensional loss increase along both N and D. Our contributions: (1) capture catastrophic overtraining (Springer et al., 2025) along D, which monotonic laws miss, and (2) extend to two dimensions including N, providing comprehensiveness that prior perturbation-aware laws lack.
>
> The Shannon-Hartley theorem provides the structural base. The log₂(1+SNR) form naturally produces U-shaped behavior when noise growth outpaces signal. This structure is not arbitrary: it is grounded in the well-established analogy between DNNs and information channels (Tishby & Zaslavsky, 2015; Shwartz-Ziv & Tishby, 2017). The power-law terms within this structure follow established scaling conventions from prior work (Kaplan et al., 2020; Hoffmann et al., 2022) and are not our novel design choices.
>
> We additionally emphasize that Shannon-Simp, a "lightweight" law with just 6 fitting constants, still outperforms all baselines (including 9-parameter ones) on unseen extrapolation data, providing strong evidence that the advantage genuinely benefits from the Shannon-Hartley structure, rather than from added parameter flexibility. We will clarify "structural analogy" vs. "rigorous derivation" in the revision.
>
> > ### W2: Relies on R^2; comparing laws with different parameter counts may not be meaningful.
>
> In the extrapolation experiments, all R² are **pooled R²**: held-out predictions across all 6 SNR levels (10–40 dB) concatenated into one flat array. This is strictly harder than per-SNR averaging. Key results: **Token extrapolation**: Shannon-Simp (6p) achieves 0.945 vs. QiD Law (9p) 0.862 at j=15. **Model extrapolation**: Shannon-Simp (6p) achieves 0.837 vs. QiD Law (9p) 0.756 at k=4. **Joint extrapolation** (most stringent): Shannon achieves 0.847 vs. QiD Law 0.761 at k=5,j=12, while Chinchilla and OpenAI laws collapse to negative R².
>
> On fairness: perturbation-aware baselines QiD Law and Law of Precision both use 9 parameters, which is same as ours. Our 6-param variant outperforms them all, making comparisons meaningful regardless of parameter count. The full Shannon (9p) = 0.847 vs. Shannon-Simp (6p) = 0.673 at joint extrapolation confirms extra parameters encode meaningful structure, not overfitting.
>
> > ### W3: Noise injection does not capture training dynamics.
>
> - We follow the same noise injection protocol as Springer et al. (2025, ICML 2025).
> - Beyond noise injection, SFT experiments (Section 4.2) capture real training dynamics: full finetuning on 6 model sizes (160M–12B) across 3 tasks with varying LR, where noise arises directly from optimization. Shannon laws achieve the highest R² on all tasks.
> - We do full finetuning (which update all model weights, scale up to 12B) for training dynamics. Full pretraining requires hundreds of GPUs beyond our capacity.
>
> > ### Q1: Practical implications? Prediction at larger scales?
>
> The extrapolation experiments validate our predictive power. Fitting on ≤6.9B models with ≤180B tokens achieves pooled R²=0.847 predicting 12B model's training up to 307B tokens, demonstrating practical utility for guiding scaling decisions.
>
> > ### Q2: Robustness across quantization schemes?
>
> We ran AWQ, bitsandbytes, and quanto on Pythia:
>
> | Method | AWQ 4-bit | bnb 4-bit | quanto 2-bit |
> |---|---|---|---|
> | **Shannon** | **0.9935** | 0.9881 | **0.9031** |
> | Shannon-Simp | 0.9837 | 0.9768 | 0.8454 |
> | Chinchilla | 0.9574 | 0.9542 | 0.0276 |
> | OpenAI | 0.9764 | 0.9679 | 0.0131 |
> | QiD Law | 0.9855 | 0.9860 | 0.8931 |
> | Law of Precision | 0.9855 | 0.9860 | 0.8932 |
> | Symmetric Law | 0.9866 | 0.9867 | 0.8531 |
> | Asymmetric Law | 0.9913 | **0.9936** | 0.8636 |
>
> The results reveal a clear pattern consistent with our GPTQ findings. Under 2-bit quanto, power laws collapse (Chinchilla: 0.028, OpenAI: 0.013) while Shannon maintains R²=0.903. This is the highest among all, outperforming QiD Law and Law of Precision.
>
> We note that the choice of quantization methods and bit-widths is not cherry-picked. Due to time constraints, we use off-the-shelf methods that are readily compatible with the Pythia architecture. Other methods such as QuIP#, AQLM, and HQQ require non-trivial engineering effort to be adapted.

---

> > ### Author Rebuttal · Reviewer_V54y · 2026-04-03
> >
> > The rebuttal provides additional experiments, particularly extrapolation results and evaluations across multiple quantization settings, which strengthen the empirical support of the proposed scaling law.
> >
> > However, the extrapolation results show that the simpler 6-parameter Shannon-Simp variant outperforms the full 9-parameter model in some settings (e.g., single-axis extrapolation). Could the authors clarify when the additional parameters are necessary, and whether this suggests that the full formulation may be over-parameterized in certain regimes?

---

> > > ### Author Response · Authors · 2026-04-04
> > >
> > > We thank reviewer V54y for the positive acknowledgment of our additional experiments and for this excellent follow-up question.
> > >
> > > The reviewer's observation is acurate and aligns with our own findings. We can offer a clear guideline on when to use each variant:
> > >
> > > **Shannon-Simp (6p) is preferred for single-axis extrapolation.** In this regime, the training data varies along only one axis (e.g., only 6 model sizes at fixed token amount). With fewer held-out points to predict, the reduced parameterization of Shannon-Simp avoids overfitting to the limited variation. For example, at j=15 there is only 1 held-out point to predict, where Shannon-Simp's efficiency is advantageous.
> > >
> > > **The full Shannon (9p) is necessary for joint extrapolation**, where one must predict across both N and D simultaneously. The joint extrapolation at k=5, j=12 makes this clear: we train on the 5 smallest models [6.9B, 2.8B, 1B, 410M, 160M] using only their first 12 token checkpoints (≤180B tokens), then predict the held-out 12B model at 4 unseen token budgets spanning **180B–307B tokens**. This is a demanding test: the law must generalize to a model 1.7× larger than anything in training, across a token range 1.7× beyond the training horizon. The full Shannon law achieves pooled R²=0.847 vs. Shannon-Simp's 0.673. This is a substantial gap. The extra parameters separately control signal scaling, data noise magnitude, and irreducible noise floor. When training data spans both axes, these terms become identifiable and capture real structure.
> > >
> > > In summary, we do not view this as over-parameterization, but as a practical consideration of data sufficiency. In real-world use cases, practitioners typically have 2 dimensional data from multiple small model sizes and shorter training runs, and wish to predict performance at a few larger-scale target configurations. This is precisely the setting of our joint extrapolation experiments. Both our fitting experiments (Tables 2–4) and extrapolation experiments consistently demonstrate that the full Shannon law achieves superior joint capability on the two-dimensional (N, D) landscape compared to all baselines. The full formulation is the appropriate choice for this scenario, enabling effective prediction well beyond the training regime.
> > >
> > > We will add this guideline to the revised manuscript. Thank you again for the constructive engagement!

---

### Official Review · Reviewer_taRX · 2026-03-24

**Soundness:** 3
**Presentation:** 3
**Significance:** 3
**Originality:** 4
**Overall Recommendation:** 5
**Confidence:** 3

**Summary:**

The authors start from highlighting the failure of existing scaling laws to explain degradation caused by, e.g., overtraining or quantization despite increased compute. They then propose a new paradigm that views LLMs as noisy channels and pretraining as channel modulation. In this information-theoretic framework, the model size maps to the bandwidth, the tokens to the signal, and the data/model/other sources of uncertainty to noise. Therefore any perturbations such as quantization induce noise and reduce the capacity. The proposed Shannon scaling law is then validated experimentally and outperforms classical scaling laws as well as perturbation-aware laws.

**Compliance With Llm Reviewing Policy:**

Affirmed.

**Key Questions For Authors:**

see weaknesses above

**Limitations:**

The paper is missing the full impact statement required by ICML.

**Strengths And Weaknesses:**

Strengths:

1. The authors provide a new paradigm and original lens that capture both what traditional scaling laws successfully captured, i.e., that compute and data increase help to an extent, and phenomena that are not explained with these traditional scaling laws, e.g., overtraining and quantization-induced perturbations.
2. The framework is theoretically motivated and elegant.
3. The scaling law is extensively validated through experiments in terms of its capacity to explain phenomena that were not captured by traditional scaling laws.
4. The proposed method successfully tracks steep curvatures of the data manifolds, demonstrating an accurate representation of the loss landscape.
5. The authors propose a formula with fewer parameters to have a comparable scaling law to parameter-efficient baselines.

Weaknesses:
1. The large number of parameters to fit (even going from 9 to 6 with the simplified formula) can induce sensitivity to parameter fit and therefore make the scaling law less precise.
2. The experiments focus on validating the scaling law rather than using to "predict" optimal scaling conditions for new scenarios. A good scaling law should be predictive and prescriptive rather than just descriptive.
3. How does test time scaling fit into this framework?

---

> ### Author Rebuttal · Authors · 2026-03-31
>
> We sincerely appreciate the reviewer's incisive evaluation! We especially value the directness and clarity of the feedback — it helped us identify the most important aspects to strengthen.
>
> > ### W1: Large number of parameters can induce sensitivity to fit.
>
> We appreciate this concern. Compared with other perturbation-aware laws (QiD Law and Law of Precision, 7–9 parameters) and classic power laws (OpenAI and Chinchilla, 4–5 parameters), our 6-parameter simplified version achieves a strong balance: it extends scaling law coverage to capture U-shaped curves across two dimensions (*N* and *D*) while maintaining competitive parameter efficiency (Table 6).
>
> To directly test stability, we ran comprehensive extrapolation experiments. All R² are **pooled R²**: held-out predictions across all 6 SNR levels (10–40 dB) concatenated into one flat array. It is harder than per-SNR averaging.
>
> **(a) Progressive Token Extrapolation.** Train on the first *j* token checkpoints, predict the rest from total 307B tokens:
>
> | Method | j=8 (Train ≤75.5B → Predict 8 pts) | j=12 (Train ≤180.4B → Predict 4 pts) | j=15 (Train ≤272.6B → Predict 1 pt) |
> |---|---|---|---|
> | **Shannon (Ours)** | 0.611 | **0.781** | **0.941** |
> | **Shannon-Simp (Ours)** | **0.768** | 0.753 | **0.945** |
> | Chinchilla | 0.197 | 0.265 | 0.352 |
> | OpenAI | 0.162 | 0.245 | 0.297 |
> | QiD Law | 0.441 | 0.766 | 0.862 |
> | Law of Precision | 0.486 | 0.632 | 0.743 |
> | Symmetric Law | 0.726 | 0.760 | 0.860 |
> | Asymmetric Law | 0.805 | 0.774 | 0.851 |
>
> **(b) Progressive Model Extrapolation.** Train on the *k* smallest models, predict larger ones:
>
> | Method | k=3 (Train [1B,410m,160m] → Predict [12B,6.9B,2.8B]) | k=4 (Train [2.8B,1B,410m,160m] → Predict [12B,6.9B]) | k=5 (Train [6.9B,2.8B,1B,410m,160m] → Predict [12B]) |
> |---|---|---|---|
> | **Shannon (Ours)** | 0.521 | 0.787 | **0.845** |
> | **Shannon-Simp (Ours)** | **0.605** | **0.837** | **0.847** |
> | Chinchilla | 0.261 | 0.726 | 0.702 |
> | OpenAI | −0.507 | −0.048 | 0.282 |
> | QiD Law | 0.470 | 0.756 | 0.721 |
> | Law of Precision | 0.294 | 0.744 | 0.713 |
> | Symmetric Law | −0.445 | 0.712 | 0.717 |
> | Asymmetric Law | −0.364 | 0.767 | 0.720 |
>
> **(c) Joint Extrapolation — the most strict test.** Extrapolate along both axes simultaneously:
>
> | Method | k=4,j=8 (Predict [12B,6.9B]×8pts) | k=5,j=12 (Predict 12B×4pts) | k=5,j=15 (Predict 12B×1pt) |
> |---|---|---|---|
> | **Shannon (Ours)** | **0.590** | **0.847** | **0.828** |
> | Shannon-Simp (Ours) | 0.494 | 0.673 | 0.715 |
> | Chinchilla | −0.146 | 0.305 | 0.439 |
> | OpenAI | −0.341 | −0.082 | −0.034 |
> | QiD Law | 0.550 | 0.761 | 0.650 |
> | Law of Precision | 0.569 | 0.708 | 0.631 |
> | Symmetric Law | 0.430 | 0.655 | 0.576 |
> | Asymmetric Law | 0.525 | 0.611 | 0.558 |
>
> **Key findings:** (1) Shannon-Simp with only 6 parameters outperforms 9-parameter QiD Law (0.837 vs. 0.756 at k=4; 0.945 vs. 0.862 at j=15)—structural advantage, not parameter advantage. (2) Shannon-Simp excels at single-axis extrapolation, while full Shannon dominates joint extrapolation (0.847 vs. 0.673), confirming extra parameters encode meaningful structure. (3) Pooled R² across 6 SNR levels further rules out overfitting. Further parameter reduction is a future goal.
>
> > ### W2: Should be predictive and prescriptive, not just descriptive.
>
> The extrapolation experiments above validate predictive power.
> Fitting on ≤6.9B models with ≤180B tokens achieves pooled R²=0.847 predicting 12B model's training up to 307B tokens, demonstrating practical utility for guiding scaling decisions.
>
> > ### W3: How does test-time scaling fit?
>
> This is a highly forward-looking question that we find very exciting. In our framework, test-time compute scaling (e.g., chain-of-thought, beam search, repeated sampling) can be interpreted as introducing **channel coding redundancy** at inference time. In communication systems, increasing coding redundancy reduces decoding error rates even over a fixed-capacity channel. Analogously, allocating more compute at inference can improve output quality without changing the model's underlying capacity. Formally, test-time compute could be modeled as an additional multiplicative factor on the effective SNR during inference, but a rigorous formulation and validation would require dedicated experiments that go beyond the scope of this paper. We will discuss this exciting direction in the revised Discussion section and plan to investigate it in future work.
>
> We hope these additional experiments have addressed the reviewer's core concerns. We would be very grateful for any further suggestions to strengthen the work!

---

### Decision · Program_Chairs · 2026-04-30

**Decision:**

Accept (regular)

**Comment:**

There is a general consensus that this paper provides good contributions on viewing LLM training through the lens of noisy channels.  The fact that this view captures non-monotonic behavior was viewed as a particular strength, and it was also appreciated that the predictions can fit observed curves better than various baselines.  Some reviewer comments suggested that some contributions could be presented more cautiously/modestly, and the reviewers should carefully take this into consideration when preparing the final version.